# GOOD BETTER BEST: SELF-MOTIVATED IMITATION LEARNING FOR NOISY DEMONSTRATIONS

## ABSTRACT

Imitation Learning (IL) aims to discover a policy by minimizing the discrepancy between the agent's behavior and expert demonstrations. However, IL is susceptible to limitations imposed by noisy demonstrations from non-expert behaviors, presenting a significant challenge due to the lack of supplementary information to assess their expertise. In this paper, we introduce Self-Motivated Imitation LEarning (SMILE), a method capable of progressively filtering out demonstrations collected by policies deemed inferior to the current policy, eliminating the need for additional information. We utilize the forward and reverse processes of Diffusion Models to emulate the shift in demonstration expertise from low to high and vice versa, thereby extracting the noise information that diffuses expertise. Then, the noise information is leveraged to predict the diffusion steps between the current policy and demonstrators, which we theoretically demonstrate its equivalence to their expertise gap. We further explain in detail how the predicted diffusion steps are applied to filter out noisy demonstrations in a self-motivated manner and provide its theoretical grounds. Through empirical evaluations on MuJoCo tasks, we demonstrate that our method is proficient in learning the expert policy amidst noisy demonstrations, and effectively filters out demonstrations with expertise inferior to the current policy.

## 1 INTRODUCTION

As a special case of sequential decision-making paradigm, Imitation Learning (Hussein et al., 2017) differs from conventional Reinforcement Learning (RL) (Sutton & Barto, 2018) by aiming to learn policies purely from offline demonstrations without relying on explicit reward signals. IL algorithms operate under the assumption that demonstrations are all drawn from expert/clean policies, hereinafter also referred to as the optimal behavior policies , and training agents by imitating them is a promising way to compensate for the unavailability of the reward function. The general goal of IL algorithms is to train an agent to generate actions that match the expert's behavior. For instance, Behavior Cloning (BC) (Bain & Sammut, 1995) aims to maximize the log-likelihood that the policy generates expert trajectories. Generative Adversarial Imitation Learning (GAIL) (Ho & Ermon, 2016) proposes to utilize Generative Adversarial Nets (GAN) (Goodfellow et al., 2014) to minimize the Jensen-Shannon (JS) divergence between the agent and the expert through a discriminator that distinguishes expert trajectories from generated trajectories.

Unfortunately, it is easier for non-expert demonstrators to obtain corrupt actions under the same state as the expert, which is mainly caused by reasons like, in many real-world tasks, operating errors made by the low expertise. This will consequentially result in non-expert/noisy demonstrations. In the presence of those demonstrations, the agent may be misguided as it cannot distinguish right from wrong, leading to compromised robustness and limited applicability. To address the issue, some approaches have introduced additional annotations, like human preference or reward signal, to indicate the expertise of the demonstrations (Brown et al., 2019; Tangkaratt et al., 2020b). By including a supervised auxiliary task of predicting the annotation, the agent can differentiate the expertise of demonstrations. However, annotations that conform to human intuition may not necessarily reflect the actual expertise of demonstrations, and in practice, such annotations may even be absent. Thus, methodologies that rely on additional annotations are susceptible to the availability and accuracy of annotations. It is preferable for the agent to automatically infer the expertise of demonstrations without requiring additional guidance.

To address the aforementioned issue, we propose Self-Motivated Imitation LEarning (SMILE). Inspired by how a human learns in a self-motivated manner, we view the agent as a beginner who can initially absorb knowledge effortlessly. As the agent becomes more experienced, it instinctively prioritizes acquiring more profound knowledge over revisiting previously learned concepts. This self-motivated framework aligns with the principles of Self-Paced Learning (SPL) (Kumar et al., 2010), encouraging models to choose samples that are more valuable for learning at each iteration.

Our proposed SMILE extends this idea by automatically identifying and filtering out demonstrations inferior to the agent, enabling the agent to keep on imitating better demonstrations for improved robustness. To accomplish this, we introduce a Policy-wise Diffusion framework, which models a Markov chain of diffusion steps. The forward diffusion process entails the gradual addition of random noise or perturbations to a policy to deteriorate its expertise. In theory, the more a policy is diffused, the worse its expertise will be. Therefore, by constructing a conditioned Q-function that considers both the noise information and the diffusion steps, we can quantify the distance between any two policies along the Markov chain and further justify its rationality, enabling the agent to exclude the samples from the dataset which are produced from a policy inferior to the current policy. The reverse process of our Policy-wise Diffusion framework is used to generate the action given a specific state. However, the long-step generation of the original diffusion model is known to incur considerable time costs for decision-making. To address this issue, we have modified the reverse process by training a policy that generates actions in a single step to approximate the outcome of the original multi-step reverse process, thereby enabling the efficient application of SMILE in sequential decision-making scenarios. Consequently, the entire policy learning process is accomplished in a self-motivated manner.

Our experimental results on MuJoCo tasks (Todorov et al., 2012) demonstrate that SMILE is robust against noisy demonstrations without the need for additional auxiliary information. Moreover, SMILE outperforms and exhibits greater interpretability than other unsupervised methods. Interestingly, SMILE achieves results comparable to methods that rely on reward signal for several tasks.

**Contributions**   Our contributions can be summarized as follows:

- We propose a Policy-wise Diffusion framework that simulates the gradual degradation of demonstration expertise through the forward process, thus enabling the agent to discern the source that corrupts the expertise of demonstrations for self-motivated learning.

- We design a metric to evaluate the superiority of one policy over another by predicting its diffusion steps and provide its theoretical underpinnings. This metric offers a solution that selects more valuable demonstrations without relying on additional annotation.

- We adapt the denoising process of the diffusion model to mitigate the long-step generative cost, thereby enabling the learned policy to be more practical for real-world applications.

## 2 PRELIMINARY

**Notations**   We formulate a standard Markov Decision Process (MDP) as a tuple $\mathcal{M} = \langle \mathcal{S}, \mathcal{A}, \mathcal{T}, \rho_0, r, \gamma, \pi \rangle$, where $\mathcal{S}$ represents state space, $\mathcal{A}$ is the action space, $\mathcal{T} : \mathcal{S} \times \mathcal{A} \times \mathcal{S} \rightarrow [0, 1]$ is the dynamic model, $\rho_0 : \mathcal{S} \rightarrow [0, 1]$ represents the distribution of the initial state $s_0$, $r : \mathcal{S} \times \mathcal{A} \rightarrow \mathbb{R}$ gives reward for a pair of state $s \sim \mathcal{S}$ and action $a \sim \mathcal{A}$, and $\pi : \mathcal{S} \times \mathcal{A} \rightarrow [0, 1]$ is the policy that selects an action at a state. The overall objective for a policy is to maximize the expectation of the cumulative discounted return $\mathcal{R}$ for $\tau = \{(s, a)_n\}_{n=0}^{N}$, which is a trajectory of length $N$. $\mathcal{R}$ is defined as: $\mathcal{R}(\tau) = \sum_{n=1}^{N} \gamma^n r(s_n, a_n)$.

**Definition 2.1.** Given two policies $\pi_1$ and $\pi_2$, their expertise are comparable to establish a partial-order relationship between the expectations they collected, denoted as:

$$\pi_1 \preceq \pi_2 \Leftrightarrow \mathbb{E}_{\tau \sim \pi_1}[\mathcal{R}(\tau)] \preceq \mathbb{E}_{\tau \sim \pi_2}[\mathcal{R}(\tau)]. \tag{1}$$

### 2.1 IMITATION LEARNING

Since the reward function is crucial to learning a value function in RL, its absence makes RL-based methods inoperable. In comparison, IL is a type of algorithm that enables an agent to learn an expert

policy without the need for ground-truth reward feedback. Based on the assumption that expert demonstrations contain the necessary information to guide a policy toward optimality, IL forces the agent's trajectories to match the expert demonstrations, which is formed as:

$$L(\pi) = \min_{\pi} \mathcal{D}(\rho^{\pi}(s,a)||\rho^E(s,a)), \tag{2}$$

where $\rho^{\pi}$ denotes the state-action density of the learned policy $\pi$ and $\rho^E$ denotes that of the expert's policy. Classic IL algorithms, such as BC (Bain & Sammut, 1995), use Mean Square Error (MSE) or Maximum likelihood estimation (MLE) as the discrepancy function $\mathcal{D}$ to train the agent. Besides, GAIL (Ho & Ermon, 2016) proposes to minimize the JS divergence of $\rho^{\pi}$ and $\rho^E$ through a discriminator that can distinguish the agent from the expert, leading to an equivalence to Inverse Reinforcement Learning (Abbeel & Ng, 2004) which learns a pseudo-reward function that gives a high reward to the expert and a low reward to the agent. Recently, Diffusion BC (Pearce et al., 2023) leverages the Diffusion Model (Sohl-Dickstein et al., 2015) to learn the transformation from the expert action distribution to a standard Gaussian distribution and then generate actions by denoising, which shows superiority in fitting the expert distribution. Unfortunately, this method assumes that demonstrations are all drawn from expert policies, which is not robust against noisy demonstrations.

## 2.2 DIFFUSION MODEL

Diffusion Model (DM) (Sohl-Dickstein et al., 2015) is a kind of generative model that generates samples by gradually transforming a pure noise (e.g., standard Gaussian noise) into a simple data distribution. It is generally divided into two opposite processes. For the $forward$ process of DM, it perturbs a data point $x_0 \sim q(x_0)$ by a certain Markovian Diffusion Kernel according to the prior simple distribution. For instance, the Gaussian Diffusion Kernel is defined as:

$$q(x_{1:T}|x_0) := \prod_{t=1}^{T} q(x_t|x_{t-1}), \quad q(x_t|x_{t-1}) := \mathcal{N}(x_t; \sqrt{1-\beta_t}x_{t-1}, \beta_t\mathbf{I}), \tag{3}$$

where $x_t$ represents the $t$-step perturbed data and $\beta_t$ comes from a time-dependent variance schedule which could either be learnable or fixed as constant (Ho et al., 2020; Nichol & Dhariwal, 2021). When $T$ is large enough, iterating this kernel would ultimately diffuse the data into the standard Gaussian distribution, i.e., $x_T \sim \mathcal{N}(0, \mathbf{I})$.

For the $reverse$ process, it aims at recovering samples step by step, starting with a pure noise from the simple distribution in the forward process, which is also called the denoising process. Likewise, take the standard Gaussian distribution as an example:

$$p_{\theta}(x_{0:T}) := p(x_T)\prod_{t=1}^{T} p_{\theta}(x_{t-1}|x_t), \quad p_{\theta}(x_{t-1}|x_t) := \mathcal{N}(x_{t-1}; \mu_{\theta}(x_t,t), \Sigma_{\theta}(x_t,t)), \tag{4}$$

where $\mu_{\theta}$ and $\Sigma_{\theta}$ are the mean and the variance of the denoising model with $\theta$ as their parameter. Finally, the generative model is summarized in the form $p_{\theta}(x_0) = \int p_{\theta}(x_{0:T})dx_{1:T}$ and it practically generates data by iterating the denoiser $p_{\theta}(x_{t-1}|x_t)$.

DM is expected to generate data subject to $q(x_0)$ after sufficient training. In detail, the original training objective is to minimize the negative log-likelihood of $p_{\theta}(x_0)$:

$$\min_{\theta} \quad \mathbb{E}_q[-\log p_{\theta}(x_0)]. \tag{5}$$

With the trained generative model $p_{\theta}$, diffusion models in every timestep of the reverse process could make a noisy sample less noisy until it is denoised to a clean sample from $q(x_0)$. However, the generative quality is critically dependent on the timestep $T$. DM generally demands a large $T$ to ensure the high quality of generated data. This time-quality dilemma remains a pain in DM.

Some research has incorporated DM into sequential decision-making problems (Janner et al., 2022; Wang et al., 2022; Pearce et al., 2023). However, they simply leverage it as a generator of actions rather than exploring the appropriate approach of diffusion and generation in the sequential decision-making paradigm. In addition, they all train their generators to fit expert distribution or use the reward signal to assist, which indicates that their models are prone to fail under noisy demonstrations.

# 3 SELF-MOTIVATED IMITATION LEARNING

This section explains the proposed approach, Self-Motivated Imitation LEarning (SMILE), which jointly performs the filtering of noisy demonstrations and the learning of the expert policy. Section 3.1 first presents a Policy-wise Diffusion framework to capture the information that triggers the deterioration of the expertise of policies. Section 3.2 then explains the modifications we made to improve the efficiency of the generative process of DM. Finally, Section 3.3 describes our self-motivated strategy for filtering out noisy demonstrations.

## 3.1 POLICY-WISE DIFFUSION

DM proposes the gradual diffusion of a sample until it finally conforms to a simple distribution, such as standard Gaussian. While this has been found effective in certain tasks, such as image and text generation, it is intuitively inconsistent with our goal to gradually reduce the expertise of policies.

Given the complexity of sources leading to demonstrator expertise corruption, we draw inspiration from existing work such as ILEED (Beliaev et al., 2022) and VILD (Tangkaratt et al., 2020b). We propose to model policies with low expertise, which lead to corrupt actions, as distorted versions of high-expertise policies. This aligns with how corrupt actions are collected in real-world tasks. Let $\mathcal{C}$ denote the corruption operator, it can be represented as $\pi^{low}(a^{low}|s) = \int \pi^{high}(a^{high}|s)\mathcal{C}(a^{low}|a^{high})\, da^{high}$. We incorporate this into the Denoising Diffusion Probabilistic Model (DDPM) (Ho et al., 2020) to gradually diminish the expertise of policies.

**Diffusion Process**   As discussed above, the purpose of the diffusion process is to gradually execute the corruption operator to perturb policies. Specifically, Gaussion is adapted as the corruption operator $\mathcal{C}$ to simulate the decline in expertise conditioned on the fixed state $s$, represented as:

$$\pi^t(a_t|s) = \int \pi^{t-1}(a_{t-1}|s)q(a_t|a_{t-1}, s)\, da_{t-1}, \quad q(a_t|a_{t-1}, s) := \mathcal{N}(a_t; a_{t-1}, \beta_t^2 \mathbf{I}), \quad (6)$$

where $\beta$ is a constant schedule. Theoretically, the larger step $t$ a policy is diffused, the less expertise of $\pi^t$ will be. For efficient training, we generate noisy policies using a closed form of Eq. 6, given as:

$$\pi^t(a_t|s) = \int \pi^0(a_0|s)q(a_t|a_0, s)\, da_0, \quad q(a_t|a_0, s) := \mathcal{N}(a_t; a_0, \sigma_t^2 \mathbf{I}), \quad (7)$$

where $\sigma_t = \sqrt{\sum_{k=1}^t \beta_k^2}$.

**Proposition 3.1.** *After Policy-wise Diffusion, it is more probable that $\pi^t$ is non-expert compared to $\pi^{t'}$, where $t' < t$.*

**Training**   Since a policy could be diffused to its corresponding less-expertise versions, we would like to capture the noise information that triggers the expertise deterioration at every step. Following DDPM (Ho et al., 2020), a neural network, $\epsilon_\theta$, is trained to predict the noise:

$$L(\theta) = \mathbb{E}_{s, a_0, t, \epsilon}[\|\epsilon - \epsilon_\theta(s, a_t, t)\|_2^2], \quad (8)$$

where $\epsilon \sim \mathcal{N}(0, \mathbf{I})$ represents the random noise applied in the reparameterization of Eq. 7 to obtain $a_t = a_0 + \sigma_t \epsilon$.

With the learned noise approximator $\epsilon_\theta$, it is allowed to **1)** generate target actions along the reverse process of DDPM and **2)** automatically identify and screen out the demonstrations collected by non-expert policies. We first discuss the generative process of SMILE and the essential modifications to adapt it to the decision-making process.

## 3.2 ONE-STEP GENERATOR

In the sequential decision-making setting, agents generate an action directly based on the given state $s$. However, the reverse process of DDPM results in the $\mathcal{O}(T)$ generation complexity that heavily compromises the decision efficiency. To make our framework more practical, we have chosen to sustain $\pi_\phi(a|s)$ as the generator of SMILE to reduce the generation complexity to $\mathcal{O}(1)$.

Then, we encourage the policy $\pi_\phi$ to directly predict the outcome of the reverse process of DDPM by utilizing the forward process posterior $q(x_{t-1}|x_t, x_0)$, which is the ground truth of denoiser $p_\theta(x_{t-1}|x_t)$ and conditioned on the generation target $x_0$. In doing so, the one-step generator is able to generate samples identical to those generated by the original multi-step generator. To wrap up in a narrative way, in SMILE, the role of the noise approximator $\epsilon_\theta$ undergoes a transformation. Its primary role shifts from serving as a denoiser for multi-step generation to guiding the one-step generator $\pi_\phi$. Thus, we adopt stop-gradient to prevent any disturbance to the original objective of $\epsilon_\theta$. The overall objective function can be summarized as follows:

$$L(\phi) = \mathbb{E}_{(s,a_0)\sim\mathcal{D}, a_0'\sim\pi_\phi, t, \epsilon}[D_{KL}(q(a_{t-1}|a_t, a_0')||p_\theta(a_{t-1}|a_t))]. \tag{9}$$

Therefore, we indirectly force $a_0'$ generated by $\pi_\phi$ to approach the ground-truth $a_0$. In the implementation, the policy will be trained with Mean Square Error to narrow the distance between the means of two distributions:

$$L(\phi) = \mathbb{E}_{(s,a_0)\sim\mathcal{D}, a_0'\sim\pi_\phi, t, \epsilon}[\|\mu_t(a_t, a_0') - \mu_t(a_t, a_t - \sigma_t\epsilon_\theta(s, a_t, t))\|^2], \tag{10}$$

where $a_t \sim q(a_t|a_0, s)$. Note that $a_t$ is equivalent to the samples obtained during the denoising process of DDPM. Figure 1 illustrates the training scheme of SMILE. This enables us to create a one-step generator for SMILE, leading to the development of a policy that can be efficiently applied in the decision-making paradigm.

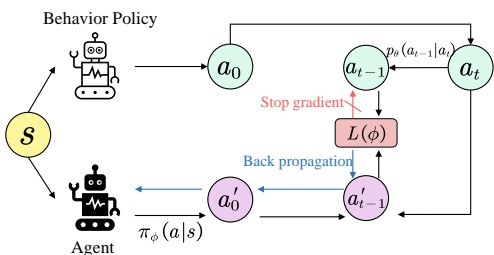

It is worth noting that Eq. 10 is theoretically equivalent to Eq. 2 as they both aim at narrowing the gap between the distributions of target and generated data. Ideally, when demonstrations are all expert, the imitator can learn the expert policy. To achieve this, we further discuss how to filter out the noisy demonstrations in a self-motivated manner.

Figure 1: One-step SMILE Generator

### 3.3 SELF-MOTIVATED FILTERING

As previously mentioned, as diffusion step $t$ increases, the expertise of the diffused policy $\pi^t$ decreases. We can use the diffusion step $t$ as a measure of the expertise gap between $\pi^t$ and $\pi^0$. Assuming that each demonstration in the dataset $\mathcal{D}$ is sampled by a specific behavior policy, we denote the current policy as $\pi_\phi$ and the behavior policy used to collect a demonstration $\tau_i$ as $\pi^{\beta_k}$ [1].

This allows us to evaluate the expertise of $\pi_\phi$ by estimating the number of diffusion steps it deviates from the behavior policy $\pi^{\beta_k}$ that contributed a group of demonstrations in dataset $\mathcal{D}$. In the consequent learning, we can select better demonstrations from the dataset without any auxiliary information by simply determining whether the current policy $\pi_\phi$ outperforms the behavior policy that produces these demonstrations. Therefore, the key to achieving such a self-motivated learning paradigm is to determine the number of diffusion steps between any two policies.

Before presenting our theoretical conclusion, we first introduce the relationship between noise and energy function. To be specific, we opt for using an expressive model, the Energy-Based Model (EBM), to represent a multimodal action distribution (Haarnoja et al., 2018; Liu et al., 2020; Song et al., 2020). For each state $s$ and action $a$, we have:

$$\pi(a|s) \propto \exp(-E(s, a)), \tag{11}$$

where $E(s, a)$ is the energy function. We then have the following proposition.

**Proposition 3.2.** *Given an action $a_t$ sampled from a diffused policy $\pi^t$ and the noise $\epsilon$, we have the gradient of $E(s, a)$ satisfying:*

$$\epsilon = -\sigma_t \nabla \log \pi(a_t|s) = \sigma_t \nabla E(s, a_t). \tag{12}$$

---

[1] We assume that the dataset $\mathcal{D}$ consists of demonstrations of diverse expertise, i.e., $\mathcal{D} = \{\tau_i\}_{i=1}^M$, and was collected using $K$ distinct behavioral policies, i.e., $\mathcal{B} = \{\pi^{\beta_k}\}_{k=1}^K$, where typically $K \ll M$.

With the trained noise approximator $\epsilon_\theta$, the gradient of energy function can be estimated. Then, we propose to compute the conditional energy function proposed by (Gao et al., 2020) with $\epsilon_\theta$ and diffusion step $t$. This energy function is proportional to the recovery likelihood of denoising a noisy sample to a clean sample. To align the RL convention, the Q-function is introduced to refer to the negative energy function. Namely in form, $p(\pi|\tilde{\pi}) \propto -E(\pi|\tilde{\pi}) = Q(\pi|\tilde{\pi})$. Using $\tilde{\pi}$ to denote the noisy policy, we obtain:

$$
\begin{aligned}
Q_{\theta,t}(\pi|\tilde{\pi}) &= \mathbb{E}_{a\sim\pi,\tilde{a}\sim\tilde{\pi}}[Q_{\theta,t}(a|\tilde{a},s)] \\
&= -\frac{1}{2\sigma_t^2}\mathbb{E}[\|a - (\tilde{a} + \sigma_t^2 \nabla Q(s,\tilde{a}))\|^2] \\
&= -\frac{1}{2\sigma_t^2}\mathbb{E}[\|a - (\tilde{a} - \sigma_t\epsilon_\theta(s,\tilde{a},t))\|^2].
\end{aligned}
\tag{13}
$$

Eq. 13 indicates how likely $\tilde{\pi}$ being denoised to $\pi$ at diffusion step $t$. In our case, $\epsilon_\theta$ is related to both $\tilde{a}$ and $t$. In implementation, to prevent the value of $Q_{\theta,t}$ from being dominated by $\frac{1}{2\sigma^2}$, we rewrite $Q_{\theta,t}(\pi|\tilde{\pi}) = -\mathbb{E}[\|a - (\tilde{a} - \sigma_t\epsilon_\theta(s,\tilde{a},t))\|^2]$. Considering a special case that $\pi$ requires 0 steps denoising to itself, we set an extra $\sigma_0 = 0$. Then we have the following propositions:

**Proposition 3.3.** *For a "clean" policy $\pi^0$ and its corresponding noisy version $\pi^t$, where $0 \leq t$, the diffusion step $t$ satisfies: $t = \arg\max_{t'} Q_{\theta,t'}(\pi^0|\pi^t)$.*

**Proposition 3.4.** *For a policy $\pi^t$ and its corresponding "cleaner" version $\pi^c$, where $c \leq t$, it is satisfied that: $0 = \arg\max_{t'} Q_{\theta,t'}(\pi^t|\pi^c)$.*

Accordingly, we can leverage $Q_{\theta,t}$ induced by $\epsilon_\theta(s, a^{\pi_\phi}, t)$ to filter out noisy demonstrations. Specifically, during the training process, we compute $t(\tau) = \arg\max_{t'\in[0,T]} \frac{1}{|\tau|} \sum_i^{|\tau|} Q_{\theta,t'}(a^{(i)}|a^{\pi_\phi}, s^{(i)})$ for each demonstration $\tau$. If $t(\tau)$ is equal to zero, it indicates that $\pi_\phi$ is of the same expertise as, or even higher than, $\pi^\beta$ for this demonstration $\tau$. Based on this criterion, we filter out noisy demonstrations from the dataset, facilitating the self-motivated learning for the policy $\pi_\phi$. This approach allows the policy to keep imitating better demonstrations until it eventually achieves best. We filter dataset at certain frequency as policy training, the overall flow is shown in Algorithm 1.

## 4 EXPERIMENTS

This section evaluates the extent to which SMILE can achieve the following goals: (1) Learn an expert policy from mixed demonstrations containing both expert and noisy demonstrations. (2) Filter out noisy demonstrations in a self-motivated manner during policy training. (3) Infer a larger diffusion step for expert demonstrations.

### 4.1 EXPERIMENTAL SETUP

**Environments** We evaluated SMILE and other baselines on various continuous-control tasks from MuJoCo (Todorov et al., 2012), such as HalfCheetah, Walker2d, and Hopper. These methods were evaluated with the cumulative reward of trajectories collected by the agent during training, where the reward was given by the ground truth reward functions predefined by the tasks.

**Datasets** Our model was trained with mixed demonstrations, which were generated with varying levels of noise. Specifically, we first trained an expert policy for each task. The expert model was then applied to collect training trajectories. To corrupt the expert's original actions, we introduce perturbations during the data collection with Gaussian noise at varying levels linearly increased from 0 to 1. We collected ten trajectories for each noise level, ultimately building a complete dataset containing 100 demonstrations in total.

**Baselines** We compared our model against two classic IL algorithms, BC (Bain & Sammut, 1995) [2] and GAIL (Ho & Ermon, 2016), as well as two other IL algorithms, RILCO (Tangkaratt et al., 2020a) and ILEED (Beliaev et al., 2022). They are also geared toward addressing the noisy demonstration

---

[2] We chose MLE as the loss function of BC in our implementation.

problem without additional annotations. RILCO proposes to divide the dataset into two parts and utilize Co-training (Blum & Mitchell, 1998) to train a pair of classifiers to label suboptimal demonstrations for each other. The label predicted is then used as the pseudo-reward to train the downstream RL algorithm ACKTR (Wu et al., 2017). ILEED optimizes a joint model that predicts the overall optimal policy and simultaneously learns to identify the suboptimality of different policies. In addition, we included COIL (Liu et al., 2021) for comparison, which is based on the return signal and proposes a curriculum strategy to select demonstrations nearest to the current policy according to the log-likelihood and filter out demonstrations with low returns to keep demonstrations better than the ones collected from the agent. To validate the effectiveness of our proposed self-motivated schema, we have also developed a variant of SMILE, SMILE w/o filtering, by simply randomly selecting a batch from the ENTIRE dataset for each iteration. To assess whether a policy reaches the optimal behavior policy, we took the average return of the expert model as the measure, where the expert model was pre-trained with SAC (Haarnoja et al., 2018) implemented in Stable Baselines3 (Raffin et al., 2019). For all the algorithms in the experiments, we measured their learning efficiency according to the accumulated transition samples they used during training.

## 4.2 COMPARISON RESULTS ON MUJOCO TASKS

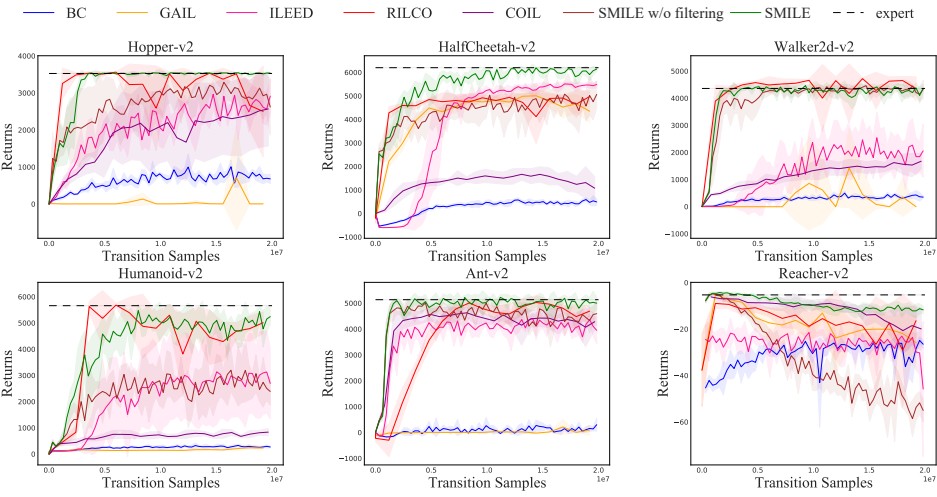

Figure 2: Results on MuJoCo tasks over 5 trials

Figure 2 provides a comparative view of the performance achieved by SMILE and the baseline methods. It is clear that SMILE, given mixed demonstrations, is able to converge to the optimal behavior policy with fewer fluctuations, outperforming other methods across most tasks.

As expected, BC and GAIL consistently demonstrated the lowest performance for most tasks since they were not specially designed for noisy demonstrations. Although ILEED and COIL outperformed BC and GAIL, they struggled to reach the expert policy. ILEED, which is trained to predict the expert policy and the expertise of demonstrations in a joint optimization setup, is prone to get trapped at local optimal solutions. In contrast, SMILE applies sequential optimization to different models. Notably, the update of $\epsilon_\theta$ is not dependent on $\pi_\phi$, thereby reducing the risk of local optima. COIL's inferior performance to our SMILE can be attributed to the inappropriate selection of demonstrations for learning. Specifically, during training COIL, we observed that it kept on choosing near-expert demonstrations, which increases the learning difficulty for the agent and contradicts the core concept of curriculum learning, which advocates progression from easy to hard tasks.

It is important to note that RILCO relies on the assumption that expert samples constitute more than half of all samples, ensuring a more accurate estimation of pseudo-labels. Thus, when this condition is unsatisfied, such as in the HalfCheetah task, RILCO fails to learn the expert policy. Moreover, we observed that although RILCO achieves expert policy on tasks that satisfy this assumption, such as Humanoid, it exhibits large variance as the number of used transition samples increases. Our conjecture is that this might be due to the misestimation of the classifier that predicts which

samples originate from noisy demonstrations because pseudo-labels provided by the dual classifier in Co-training could be misleading, which will further cause the instability of pseudo-reward and the approximation of the value function in the downstream RL algorithm. In contrast, SMILE exhibits a relatively stable training process with low variance.

The significant performance disparity between SMILE and SMILE w/o filtering for most tasks demonstrates the importance of our proposed self-motivated learning schema. It is worth noting that SMILE w/o filtering also shows competitive performance for most tasks in comparison with other baselines. This validates the excellent capability of DDPM to fit the policy distribution in the sequential decision-making paradigm.

### 4.3 INTERPRETABILITY OF SELF-MOTIVATED LEARNING SCHEME

This section primarily illustrates how the self-motivated filtering module works during training by examining the expertise of remaining demonstrations after filtering. We expect these demonstrations to be of more expertise than those generated by the current policy. In our experiments, SMILE activates the filtering module every 2500 iterations. To ensure the filtered dataset is not empty, we set a threshold of a minimum of 10 demonstrations. Meeting this threshold deactivates the filtering module.

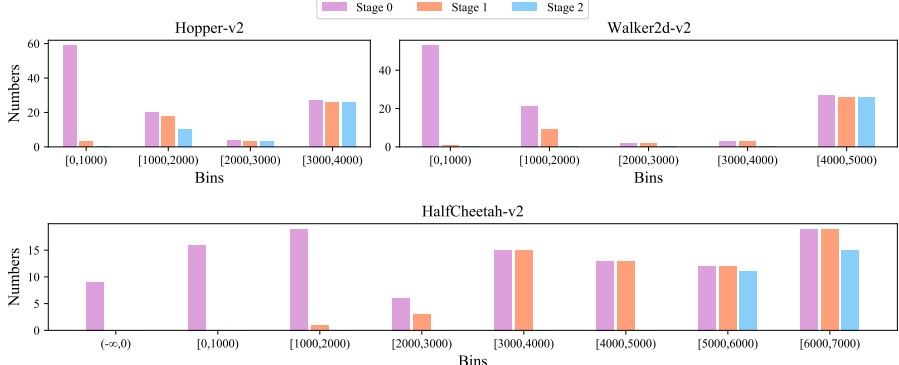

Figure 3: Histogram of remaining demonstrations regarding their Returns at different training stages

**Observation 1: Effectiveness of filtering noisy demonstrations**  Figure 3 illustrates the histogram of demonstrations for Hopper, Walker, and HalfCheetah at three different stages during the entire training process. Stage 0 denotes the initial stage of the training process before any filtering is conducted. Stage 2 commences when the current policy achieves expert and continues until convergence. Therefore, stage 1 encapsulates the period between stages 0 and 2. The x-axis represents the bins of the Return for all demonstrations. The colored bars indicate the number of demonstrations at different stages on the y-axis. We observe that noisy demonstrations predominate in the original dataset for Hopper and Walker2d, as indicated by taller purple bars in bins with smaller Returns. In contrast, the expertise in HalfCheetah is relatively balanced. Over the training period, noisy demonstrations are largely eliminated, specifically in Walker2d and HalfCheetah, such that the remaining demonstrations at stage 2 are almost exclusively from experts. The behavior of our proposed self-motivated filtering, which is based on predicted diffusion steps, is consistent with filtering based on demonstration returns as known by the oracle. This observation could explain the robustness of our model against noisy demonstrations and its ability to learn the expert policy.

Table 1: Average diffusion steps predicted for every bin on HalfCheetah

| Average Return(std) | (-∞,0] | (0,1000] | (1000,2000] | (2000,3000] | (3000,4000] | (4000,5000] | (5000,6000] | (6000,7000] |
|---|---|---|---|---|---|---|---|---|
| 2458.26($\pm$1537.13) | 0.00 | 0.26 | 0.82 | 2.16 | 3.13 | 3.00 | 3.00 | 3.00 |
| 4354.33($\pm$1395.95) | N/A | N/A | N/A | 0.66 | 2.00 | 2.69 | 2.25 | 2.52 |
| 5484.62($\pm$1194.35) | N/A | N/A | N/A | N/A | 1.38 | 2.23 | 2.16 | 2.05 |
| 6182.89($\pm$108.83) | N/A | N/A | N/A | N/A | 0.00 | 1.54 | 1.75 | 1.63 |

**Observation 2: Alignment between diffusion steps and expertise gap**  To further elucidate the interpretability of the self-motivated filtering module, which is based on predicted diffusion steps, Table 1 presents the averaged predicted diffusion steps for demonstrations grouped in each return bin in the HalfCheetah task as the current policy evolves. The leftmost column displays the average return achieved by the current policy, along with its standard deviation over 10 evaluations. $N/A$ means that all demonstrations in the corresponding bin have been filtered out. The observation indicates that demonstrations inferior to the agent are predicted with smaller diffusion steps than those superior to the agent. This reflects that SMILE can identify the expertise gap between demonstrations and current policy. Although the average predicted diffusion steps do not strictly increase along with the bins of Returns [3], this inconsistency does not impact the selection of demonstrations for filtering, as demonstrations with high returns are always preserved. Our conjecture is that the discrepancy may arise from training errors stemming from the concurrent training of $\epsilon_\theta$ and the policy. As the agent is trained to gradually approach the expert, we observe a decrease in the average predicted diffusion steps corresponding to each bin. This suggests that the predicted diffusion steps effectively capture the trend of the shrinking expertise gap relative to all remaining demonstrations. In conclusion, the predicted diffusion steps effectively represent the expertise gap, which confirms the interpretability of the filtering module of SMILE and assures the safe filtering out of noisy demonstrations.

## 5  RELATED WORK

In this section, we introduce several additional methods that are not employed as baselines due to reasons such as the absence of annotations. These algorithms are also designed to address the challenge of noisy demonstrations.

Partially labeling the data with the confidence score, which indicates the probability for one demonstration to be an expert, IC-GAIL (Wu et al., 2019) extends GAIL by first training a classifier to estimate the likelihood of a demonstration originating from an expert. Meanwhile, VILD (Tangkaratt et al., 2020b) predicts the expertise of demonstrations when the identity of corresponding demonstrators is known. T-REX (Brown et al., 2019), which is an Inverse Reinforcement Learning (IRL) (Ng et al., 2000; Abbeel & Ng, 2004) algorithm, utilizes rankings of trajectories to train a pseudo-reward function that gives higher reward for trajectories on the top of the ranking list. Although these methods are capable of guiding agents in learning an expert policy, they heavily rely on additional information that may be absent in practical scenarios. There are also IRL algorithms without annotations are proposed, like DREX(Brown et al., 2020) and SSRR(Chen et al., 2021), using noise to learn reward.

## 6  DISCUSSION

**Limitations and Future Work**  There remains some possible modifications of SMILE in the future. For example, it is currently only applicable in continuous action spaces. The challenge arises from our form of diffusion being based on Gaussian noise, which, when added, can inadvertently diffuse a poor action to a better one in a finite action space. Therefore, future work will investigate different forms of diffusion to identify the most appropriate and meaningful approach to degrade the expertise of demonstrations in both continuous and discrete action spaces. Moreover, we acknowledge the underutilization of order relations. Although SMILE can distinguish higher-expertise demonstrations from all saved demonstrations, we have yet to determine an effective method for leveraging the relationship between mediocre, good yet non-expert, and expert demonstrations. Furthermore, we will explore integrating state feature extraction methods to improve the applications of SMILE.

**Conclusion**  Traditional IL methods are not robust in learning against noisy demonstrations, often relying on additional annotation or lacking interpretability. This paper introduces a self-motivated IL-based method, SMILE, which can automatically identify noisy demonstrations without any additional information through Policy-wise Diffusion. Furthermore, we adapt both the diffusion and generative processes of DDPM to accommodate sequential decision problems. Empirical results validate the effectiveness of SMILE and demonstrate its interpretability, showing that SMILE is a robust and comprehensible solution to the problem of noisy demonstrations.

---

[3]For instance, the predicted diffusion steps of demonstrations in the $(5000, 6000]$ bin sometimes exceed those in the $(6000, 7000]$ bin.

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

# A APPENDIX

The Appendix provides supplementary information, including proofs and additional experiments.

## A.1 PROOFS

### A.1.1 DERIVATION OF EQ. 7

*Proof.* We first illustrate how we get the closed form of Policy-wise Diffusion. By reparameterizing the original diffusion kernel in Eq. 6, we demonstrate the actions from timestep $t$ to $0$ according to the chain-style derivation:

$$
\begin{aligned}
a_t &= a_{t-1} + \beta_{t-1}\epsilon_{t-1} \\
&= a_{t-2} + \beta_{t-2}\epsilon_{t-2} + \beta_{t-1}\epsilon_{t-1} \\
&= ... \\
&= a_0 + \underbrace{\beta_0\epsilon_0 + \beta_1\epsilon_1 + ... + \beta_{t-1}\epsilon_{t-1}}_{merge\ \ standard\ \ Gaussians} \\
&= a_0 + \sqrt{\beta_0^2 + \beta_1^2 + .., + \beta_{t-1}^2}\,\epsilon \\
&= a_0 + \sigma_t\epsilon
\end{aligned}
\tag{14}
$$

With the notation that $\sigma_t = \sqrt{\sum_{k=1}^{t}\beta_k^2}$, it is obvious that $a_t$ could be represented by the reparameterization of samples subject to a Gaussian with mean $a_0$: $a_t \sim \mathcal{N}(a_0, \sigma_t^2\mathbf{I})$. Therefore, we can rewrite the forward kernel of Policy-wise Diffusion as $q(a_t|a_0, s) = \mathcal{N}(a_0, \sigma_t^2\mathbf{I})$

∎

### A.1.2 DERIVATION OF EQ. 13

*Proof.* Given the detailed form of Eq. 11:

$$
\pi(a|s) = \frac{exp(Q(s,a))}{Z}
\tag{15}
$$

where $Z$ is the partition function and $Q(s,a) = -E(s,a)$.

Then, we can derive the conditional EBM of $x$ given its noisy version that $\tilde{a} = a + \sigma_t\epsilon$ as:

$$
\begin{aligned}
q(a|\tilde{a}, s) &= \frac{\pi(a|s)q(\tilde{a}|a, s)}{\pi(\tilde{a}|s)} \\
&= \frac{\frac{1}{Z}exp(Q(s,a))\frac{1}{(2\pi\sigma_t^2)^{\frac{n}{2}}}exp(-\frac{1}{2\sigma_t^2}\|\tilde{a} - a\|^2)}{\pi(\tilde{a}|s)} \\
&= \frac{exp(Q(s,a) - \frac{1}{2\sigma_t^2}\|\tilde{a} - a\|^2)}{\tilde{Z}}
\end{aligned}
\tag{16}
$$

where we absorb all the terms that are irrelevant to $a$ as $\tilde{Z}$.

Therefore, the conditioned energy function can be described as:

$$
\begin{aligned}
-E(a|\tilde{a}, s) &= Q(s,a) - \frac{1}{2\sigma_t^2}\|\tilde{a} - a\|^2 \\
&\approx Q(s, \tilde{a}) + <\nabla Q(s, \tilde{a}), a - \tilde{a}> - \frac{1}{2\sigma_t^2}\|\tilde{a} - a\|^2 \\
&= -\frac{1}{2\sigma_t^2}[\|a - (\tilde{a} + \sigma_t^2\nabla Q(s, \tilde{a}))\|^2] + C \\
&= Q(a|\tilde{a}, s)
\end{aligned}
\tag{17}
$$

∎

### A.1.3 PROOF OF PROPOSITION 3.1

**Expert Policy**    For the expert policy $\pi_{exp}(a^*|s)$ , we diffuse it by $q(\tilde{a}|a^*, s)$ at every simulation time step during the interaction according to Eq. 7. Suppose any action that deviates $a^*$ less than $\alpha$, which corresponds to the interval $[a^* - \alpha, a^* + \alpha]$, is expert, we can then present the probability that the diffused policy $\tilde{\pi}$ is non-expert. We first introduce the situation that $\tilde{\pi}$ is expert:

$$
\begin{aligned}
p_{exp}(\tilde{\pi}) &= \mathbb{E}_{\tilde{\pi}}[p_{exp}(\tilde{a})] \\
&= \mathbb{E}_{\tilde{\pi}}[p(|\tilde{a} - a^*| \leq \alpha)] \\
&= \mathbb{E}_{\tilde{\pi}}\big[\frac{1}{\sqrt{2\pi}\sigma} \int_{a^*-\alpha}^{a^*+\alpha} e^{-\frac{(\tilde{a}-a^*)^2}{2\sigma^2}} d\tilde{a}\big] \\
&= \mathbb{E}_{\tilde{\pi}}\big[\frac{1}{\sqrt{\pi}} \int_{-\frac{\alpha}{\sqrt{2}\sigma}}^{\frac{\alpha}{\sqrt{2}\sigma}} e^{-(\frac{\tilde{a}-a^*}{\sqrt{2}\sigma})^2} d(\frac{\tilde{a}-a^*}{\sqrt{2}\sigma})\big] \\
&= \mathbb{E}_{\tilde{\pi}}\big[\frac{1}{\sqrt{\pi}} \int_{-\frac{\alpha}{\sqrt{2}\sigma}}^{\frac{\alpha}{\sqrt{2}\sigma}} e^{-y^2} dy\big] \\
&= \mathbb{E}_{\tilde{\pi}}\big[\frac{1}{\sqrt{\pi}} \frac{\sqrt{\pi}}{2} erf(y)|_{-\frac{\alpha}{\sqrt{2}\sigma}}^{\frac{\alpha}{\sqrt{2}\sigma}}\big] \\
&= \mathbb{E}_{\tilde{\pi}}[erf(\frac{\alpha}{\sqrt{2}\sigma})]
\end{aligned}
\tag{18}
$$

where $erf(x)$ is the Gaussian Error Function whose range is from -1 to 1, and it is positively related to $x$. Therefore, we can then present $p_{non}(\tilde{\pi})$ as :

$$
\begin{aligned}
p_{non}(\tilde{\pi}) &= 1 - p_{exp}(\tilde{\pi}) \\
&= 1 - \mathbb{E}_{\tilde{\pi}}[erf(\frac{\alpha}{\sqrt{2}\sigma})]
\end{aligned}
\tag{19}
$$

Obviously, as $\sigma$ grows from 0 to $+\infty$, $\frac{\alpha}{\sqrt{2}\sigma}$ will shrink from $+\infty$ to 0, which results in that when $\sigma$ is close to zero, $p_{non}(\tilde{\pi})$ is also near zero, otherwise when $\sigma$ is large enough, $p_{non}(\tilde{\pi})$ is close to 1. Since the deviation is up to $\sigma$, it indicates that the larger the noise level is, the worse a diffused policy $\tilde{\pi}$ could be compared to the expert policy $\pi_{exp}$ . According to this, it is proved that Policy-wise Diffusion is able to degrade the expertise of expert policies by raising the deviation from $\pi_{exp}$.

**Non-expert Policy**    For any action $a$ generated by a non-expert policy $\pi$ given a state $s$, whose expertise is compromised, we analyze whether Policy-wise Diffusion can still decrease the expertise of non-expert policy. Consistent with the main text, we view non-expert policies as the corrupt version of expert policies. Furthermore, we diffuse non-expert $\pi$ to a noisier policy $\tilde{\pi}$. If $p_{non}(\pi) < p_{non}(\tilde{\pi})$, then it is proved that adding the noise also could corrupt the expertise of non-expert policies.

We illustrate this by first reparameterizing the diffusion of $a$ and $\tilde{a}$, which leads to $a = a^* + \sigma_1\epsilon_1$ and $\tilde{a} = a + \sigma_2\epsilon_2$. With this, we can further derive:

$$
\begin{aligned}
\tilde{a} &= a + \sigma_2\epsilon_2 \\
&= a^* + \sigma_1\epsilon_1 + \sigma_2\epsilon_2 \\
&= a^* + \sqrt{\sigma_1^2 + \sigma_2^2}\epsilon
\end{aligned}
\tag{20}
$$

Based on this, we get a closed form of the distribution of $\tilde{a} \sim \mathcal{N}(a^*, \sigma_1^2 + \sigma_2^2)$. Therefore, the diffusion of the policies are summarized as $\pi(a|s) = \int \pi_{exp}(a^*|s)\mathcal{N}(a^*, \sigma_1^2)da^*$ and $\tilde{\pi}(\tilde{a}|s) = \int \pi_{exp}(a^*|s)\mathcal{N}(a^*, \sigma_1^2 + \sigma_2^2)da^*$. To wrap up, the deviation of $\pi$ and $\tilde{\pi}$ are $\sigma_1$ and $\sqrt{\sigma_1^2 + \sigma_2^2}$ respectively. Further, as discussed above, the deviation and the probability of sampling a noisy action are positively related. Because of $\sqrt{\sigma_1^2 + \sigma_2^2} > \sigma_1$, we can easily come to the conclusion that $p_{non}(\pi) < p_{non}(\tilde{\pi})$.

A.1.4 PROOF OF PROPOSITION 3.2

*Proof.* We first prove the equivalence between the noise $\epsilon$ and the gradient of the forward kernel of Policy-wise Diffusion $q(a_t|a_0, s)$. With the probability density function of Gaussian, we get:

$$q(a_t|a_0, s) = \frac{1}{\sqrt{2\pi}\sigma_t} exp(-\frac{\|a_t - a_0\|_2^2}{2\sigma_t^2}) \tag{21}$$

Then, we take the log function for both the left- and right-hand sides:

$$\log q(a_t|a_0, s) = C - \frac{\|a_t - a_0\|_2^2}{2\sigma_t^2} \tag{22}$$

where $C$ is a constant that has nothing to do with $a_t$. Next, we can discover the gradient of $a_t$:

$$\begin{aligned} \nabla_{a_t} \log q(a_t|a_0, s) &= -\frac{2(a_t - a_0)}{2\sigma_t^2} \\ &= -\frac{\sigma_t \epsilon}{\sigma_t^2} \\ &= -\frac{\epsilon}{\sigma_t} \end{aligned} \tag{23}$$

It is obvious that $\epsilon = -\sigma_t \nabla_{a_t} \log q(a_t|a_0, s)$. Based on this, we further illustrate the relationship between the gradient of $q(a_t|a_0, s)$ and $\pi(a_t|s)$:

$$\begin{aligned} \nabla_{a_t} \log \pi(a_t|s) &= \frac{\nabla_{a_t} \pi(a_t|s)}{\pi(a_t|s)} \\ &= \frac{\int \frac{1}{\sigma_t^2} q(a_t|a_0, s)(a_0 - a_t)\pi(a_0|s)da_0}{\pi(a_t|s)} \\ &= \frac{1}{\sigma_t^2} \int q_s(a_0|a_t)(a_0 - a_t)da_0 \\ &= \frac{1}{\sigma_t^2} (\mathbb{E}_{q_s(a_0|a_t)}[a_0] - a_t) \end{aligned} \tag{24}$$

To approximate $\mathbb{E}_{q_s(a_0|a_t)}[a_0]$, we introduce Tweedie's FormulaEfron (2011). It indicates that for a sample subject to Gaussian, $x \sim \mathcal{N}(\mu, \Sigma)$, the approximation of its mean can be presented by $\mathbb{E}[\mu|x] = x + \Sigma\nabla_x \log p(x)$. According to this, we can estimate $a_0$ by Tweedie's Formula like the following:

$$\begin{aligned} \nabla_{a_t} \log \pi(a_t|s) &= \frac{1}{\sigma_t^2} (a_t + \sigma_t^2 \nabla_{a_t} \log q(a_t|a_0, s) - a_t) \\ &= \nabla_{a_t} \log q(a_t|a_0, s) \end{aligned} \tag{25}$$

Therefore, we can discover that the gradient of $\pi(a_t|s)$ equals the gradient of $\log q(a_t|a_0, s)$. Moreover, we can simply get that $\epsilon = -\sigma_t \nabla_{a_t} \log q(a_t|a_0, s) = -\sigma_t \nabla_{a_t} \log \pi(a_t|s)$. Besides, since we model policy $\pi$ as EBM, as mentioned in Eq. 11, Proposition 3.2 can be proved.

∎

A.1.5 PROOF OF PROPOSITION 3.3

*Proof.* We will prove that only when $t'$ is the ground-truth diffusion step $t$, the value of $Q_{\theta, t'}(\pi_0|\pi_t)$ will reach the upper bound.

$$
\begin{aligned}
Q_{\theta,t'}(\pi_0|\pi_t) &= \mathbb{E}[Q_{\theta,t'}(a_0|a_t,s)] \\
&= -\mathbb{E}[\|a_0 - (a_t - \sigma_{t'}\epsilon_\theta(s,a_t,t'))\|^2] \\
&= -\mathbb{E}[\|a_0 - (a_t - \sigma_t\epsilon_\theta(s,a_t,t) + \sigma_t\epsilon_\theta(s,a_t,t) - \sigma_{t'}\epsilon_\theta(s,a_t,t'))\|^2] \\
&= -\mathbb{E}[\|\underbrace{a_0 - (a_t - \sigma_t\epsilon_\theta(s,a_t,t))}_{T_1} + \underbrace{(\sigma_{t'}\epsilon_\theta(s,a_t,t') - \sigma_t\epsilon_\theta(s,a_t,t))}_{T_2}\|^2] \quad (26)\\
&= \mathbb{E}[Q_{\theta,t}(a_0|a_t,s)] - 2\mathbb{E}[T_1 * T_2] - \mathbb{E}[T_2^2] \\
&= Q_{\theta,t}(\pi_0|\pi_t) - 2\mathbb{E}[T_1 * T_2] - \mathbb{E}[T_2^2]
\end{aligned}
$$

When $\epsilon_\theta$ is sufficiently trained, the term $T_1$ will be infinitely close to 0 for $a_0 - a_t + \sigma_t\epsilon_\theta(s,a_t,t) = \sigma_t\epsilon_\theta(s,a_t,t) - \sigma_t\epsilon$, where $\epsilon$ is the ground-truth of $\epsilon_\theta(s,a_t,t)$. Hence, $Q_{\theta,t}(\pi_0|\pi_t)$ and $Q_{\theta,t'}(\pi_0|\pi_t)$ will be infinitely close to 0 and $-\mathbb{E}[T_2^2]$, respectively.

While for $T_2$, it is easy to say that :

$$
T_2 \begin{cases} = 0, when \quad t' = t \\ > 0, \quad otherwise \end{cases} \quad (27)
$$

Then, we can find that:

$$
Q_{\theta,t'}(\pi_0|\pi_t) \begin{cases} = 0, when \quad t' = t \\ < 0, \quad otherwise \end{cases} \quad (28)
$$

Therefore, it can be proved that $t = \arg\max_{t'} Q_{\theta,t'}(\pi^0|\pi^t)$.

∎

### A.1.6    PROOF OF PROPOSITION 3.4

*Proof.* From what is declared above, we can further discover that for a pair of policies $(\pi^0, \pi^t)$, the numerical value of $Q_{\theta,t'}(\pi^0|\pi^t)$ will decrease as $t'$ going far away from $t$.

According to Eq. 26, when $\epsilon_\theta$ is sufficiently trained, we have:

$$
Q_{\theta,t'}(\pi^0|\pi^t) = Q_{\theta,t}(\pi^0|\pi^t) - \mathbb{E}[\|\sigma_{t'}\epsilon_\theta(s,a_t,t') - \sigma_t\epsilon_\theta(s,a_t,t)\|^2]. \quad (29)
$$

We then examine the last term on the right-hand side. Let $t' = t - k$, where $0 < k < t$, $\epsilon_\theta$ will denoise $a_t$ for $t - k$ steps to a cleaner action $a_k$, which leads to:

$$
\begin{aligned}
&\mathbb{E}[\|\sigma_{t'}\epsilon_\theta(s,a_t,t') - \sigma_t\epsilon_\theta(s,a_t,t)\|^2] \\
&= \mathbb{E}[\|(a_t - a_k) - (a_t - a_0)\|^2] \\
&= \mathbb{E}[\|a_0 - a_k\|^2].
\end{aligned} \quad (30)
$$

Along the same path, when $t' = t - s$, where $k < s < t$, we have $\mathbb{E}[\|\sigma_{t'}\epsilon_\theta(s,a_t,t') - \sigma_t\epsilon_\theta(s,a_t,t)\|^2] = \mathbb{E}[\|a_0 - a_s\|^2]$.

Since $k < s$, it means that $a_s$ is deviated more from $a_0$, and it is easy to tell that $\mathbb{E}[\|a_0 - a_k\|^2] < \mathbb{E}[\|a_0 - a_s\|^2]$. Therefore, $Q_{\theta,t-k}(\pi^0|\pi^t) > Q_{\theta,t-s}(\pi^0|\pi^t)$. Meanwhile, it is prone to validate its dual conclusion that $Q_{\theta,t+k}(\pi^0|\pi^t) > Q_{\theta,t+s}(\pi^0|\pi^t)$. To wrap up, the farther $t'$ is away from $t$, the less the value of $Q_{\theta,t'}$ will be.

Then, we could say, $Q_{\theta,t'}(\pi|\tilde{\pi})$ reflects the negative distance between $\pi$ and the policy which is denoised $t'$ steps from $\tilde{\pi}$. For $Q_{\theta,t'}(\pi^t|\pi^c)$, denoising any steps of $\pi^c$ will only make it farther away from $\pi^t$. Therefore, denoising 0 steps will make the largest value of $Q_{\theta,t'}(\pi^t|\pi^c)$.

∎

---

**Algorithm 1** SMILE Pseudocode, PyTorch-like

```
# ema_denoiser, denoiser: mlp
# ema_policy, policy: mlp
# denoiser_optimize_every: How many times the denoiser will be optimized in one step
# policy_optimize_every: How many times the denoiser will be optimized in one step
# update_ema_every: How many steps to perform ema
# filter_dataset_every: How many steps to perform filter
def train(denoiser, policy, dataset, batch_size):
    step = 0
    while step*batch_size < 2e7:
        ### sample states and actions from datasets for training
        state, action = dataset.sample(batch_size)

        ### optimize models
        for i in range(denoiser_optimize_every):
            denoise_loss = denoiser.denoise_loss(state, action)
            (denoise_loss / (denoiser_optimize_every)).backward()
        optimize(denoiser)
        for i in range(policy_optimize_every):
            policy_loss = policy.agent_loss(state, action)
            (policy_loss / (policy_optimize_every)).backward()
        optimize(policy)

        ### step ema
        if step % update_ema_every == 0:
            step_ema(ema_denoiser, denoiser)
            step_ema(ema_policy, policy)

        ### filter
        if step % filter_dataset_every == 0 :
            filter_dataset(ema_denoiser,ema_policy,dataset)

        step += 1
```

---

### A.2 PSEUDOCODE

The pseudocode of the whole training framework and the filter module of SMILE are placed in Algorithm 1 and Algorithm 2 respectively.

Algorithm 1 primarily outlines the synchronized training of $\epsilon_\theta$ and $\pi_\phi$, and how the self-motivated filtering module interleaves with the policy training process. Specifically, at each training iteration, we first train the noise approximator $\epsilon_\theta$ according to Eq. 8 and then the $\pi_\phi$ according to Eq. 10. Typically, $\epsilon_\theta$ undergoes more updates per iteration than $\pi_\phi$ to prevent disruptions in policy learning caused by errors from an incompletely trained noise approximator. Additionally, at certain training iterations interval, the self-motivated module is activated to filter out demonstrations in the dataset that exhibit less expertise compared to on-training policy $\pi_\phi$. By integrating these modules described earlier as this way, we construct the overall workflow of SMILE.

### A.3 FURTHER RESULTS

This section provides some empirical observations about SMILE that cannot be placed in the main text due to the limited space.

**Hyperparameters**  In SMILE, we set the diffusion step at 10 and $\beta$ as constants that linearly increase from 0.05 to 0.6. This ensures a gradual rather than abrupt Policy-wise Diffusion, which is beneficial for the denoiser to capture more fine-grained expertise information. In addition, we found that pragmatically setting the filter threshold at 1, which means filtering out $\tau$ when $t(\tau) \leq 1$, accelerates the filtering of low-expertise demonstrations. It is noteworthy that the traditional generative process of DDPM is dependent on a fully trained $\epsilon_\theta$. However, our model concurrently trains $\epsilon_\theta$ and $\pi_\phi$, which could potentially corrupt policy training due to the fluctuating $\epsilon_\theta$. To address this issue, during the training of the one-step generator, we updated $\epsilon_\theta$ with 10 additional gradient steps in one iteration compared to $\pi_\phi$ to ensure that the agent is guided correctly and steadily. We also applied the Exponential Moving Average (EMA) during the update of learnable parameters to improve the stability. For each iteration, a batch of 128 state-action pairs $(s, a)$ was sampled for training. The learning rate of both two models was set to $1e-3$. Besides, we found that $l1-norm$ performs better for the training of $\epsilon_\theta$. For $\pi_\phi$, we retained MSE as its training objective, as described in Section 3.2. It is necessary to emphasize that SMILE shares the same set of hyperparameters across all tasks.

**Algorithm 2** Filtering of SMILE Pseudocode, PyTorch-like

```python
def filter_demos(dataset, Q):
    '''
    (s,a,t): corresponds to state, action, terminal respectively
    max_demo_len: the longest number of a demonstration
    '''
    start, demos, num_demos = 0, [], 0

    for (s,a,t) in dataset:
        if t == True or i + 1 - start >= max_demo_len:
            argmax_t = torch.argmax(torch.mean(Q[start:i + 1]))

            if argmax_t > 0 :
                demos += data[start:i + 1]
                num_demos+=1

            start = i + 1

    return demos, num_demos

def filter_dataset(denoiser, policy, dataset):
    '''
    compute_Q(states,a_policy,actions,t): return $Q_t(actions|a_policy)$
    threshold: The least number of trajectories in the dataset
    stop_filtering: an indicator whether keep filtering
    '''
    states, actions = dataset.sample_all()
    a_policy = policy.sample_action(states)

    Q = [[] for _ in range(denoiser.num_diffusion_steps)]
    for t in range(denoiser.num_diffusion_steps):
        Q[t] = denoiser.compute_Q(states, a_policy, actions, t)

    # update dataset
    demos, num_demos = filter_demos(datasets, Q)
    if num_trajs >= threshold:
        update_dataset(dataset, demos)
    else:
        stop_filtering=True
```

Table 2: Comparison to Naive Self-Paced Learning

|  | Hopper-v2 | Walker2d-v2 | HalfCheetah-v2 | Ant-v2 | Humanoid-v2 | Reacher-v2 |
|---|---|---|---|---|---|---|
| naive SPL | 280.38($\pm$ 136.22) | 322.55($\pm$ 174.38) | 2750.84($\pm$ 558.91) | -131.87($\pm$ 147.55) | 156.11($\pm$ 55.70) | -128.09($\pm$ 49.06) |
| SMILE | 3523.80($\pm$ 14.91) | 4417.73($\pm$ 63.06) | 6009.01($\pm$ 105.21) | 5252.06($\pm$ 145.96) | 5661.81($\pm$ 273.83) | -8.57($\pm$ 4.59) |

### A.3.1 COMPARISON TO NAIVE SELF-PACED LEARNING

It is worth highlighting the distinctiveness of SMILE in comparison with traditional SPL. As a regression approach, the purpose of SPL selecting easier samples, which are measured as samples inducing lower fitting loss, is to alleviate the local optima. In comparison, SMILE aims at sidestepping noisy ones to improve robustness. Moreover, for noisy demonstrations, simply leveraging the measure in SPL cannot guarantee an effective sample selection, as demonstrations with lower fitting loss do not necessarily correspond to noisy demonstrations. Therefore, SMILE develops a novel self-motivated filtering module to automatically discern the expertise of demonstrations based on diffusion steps, keeping on imitating "better" demonstrations and excluding "worse" ones during the training process. In other words, SMILE brought a brand new view to self-paced learning, which is about how to distinguish "easy" samples (less-expertise demonstrations) from the "hard" ones (more-expertise demonstrations) in decision-making situation.

Note that SMILE achieves self-paced learning by modeling the transformation of policy expertise using the Diffusion Model. This allows the model to infer the expertise of demonstrations based on the diffusion steps and to exclude those of low expertise. This is fundamentally different from the naive SPL approach, which recognizes easy samples according to the loss function. Therefore, we further illustrate comparisons between SMILE and the naive SPL to emphasize the effectiveness of our model.

First, let's clarify the setting of the naive SPL. It considers samples with lower loss to be those the agent has already learned, and they will be excluded if their loss falls below a certain threshold.

Once the dataset is emptied, the training will be stopped. To guarantee fairness, we evaluated the performance of SMILE at the same number of iterations. As shown in Table 2, the strategy of naive SPL is unable to distinguish noisy demonstrations, which results in poor performance. In contrast, SMILE successfully excludes noisy demonstrations, thus achieving better performance.

### A.3.2 COMPARISON TO NAIVE REVERSE PROCESS

In this part, we investigate whether our one-step generator improves decision-making efficiency and demonstrate the comparison between the one-step generator and multi-step generation of a naive reverse process.

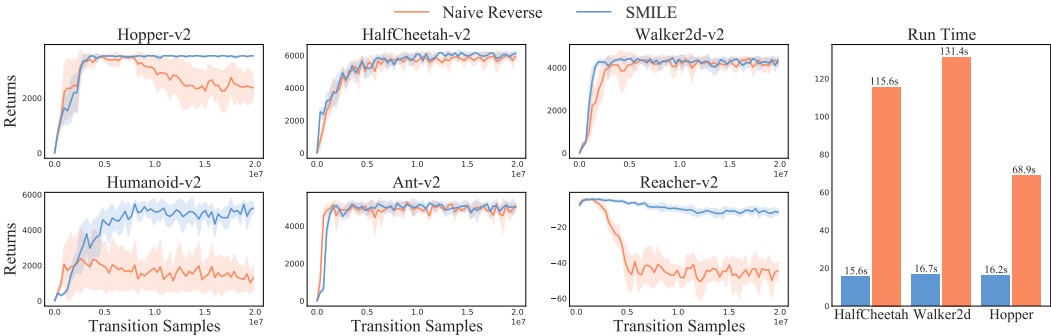

Figure 4: Learning Curve and Time Costs over 10 interactions

**Time Costs**   We first illustrate the improved decision-making efficiency gained with the one-step generator. We compare SMILE, which uses the one-step generator, to a naive reverse approach that relies on multi-step generation, recursively denoising a sample using $\epsilon_\theta$ at each diffusion step. The evaluation examines the time costs incurred across ten interactions for three tasks. To enable a fair comparison despite varying interaction lengths, we scale the times to a consistent 1000 steps. As shown in the rightmost histogram in Figure 4, the average time costs for the naive reverse are much higher than for SMILE, indicating poor decision-making efficiency. Furthermore, for HalfCheetah and Walker2d, the naive reverse's time costs are approximately ten times higher than SMILE's, suggesting it requires exactly ten more steps per decision compared to SMILE.

**Learning effect**   As all learning curves shown in Figure 4, on most tasks like HalfCheetah and Walker2d, the learning effect between naive reverse and SMILE has no obvious differences, which illustrates that the one-step generator is able to improve the decision-making efficiency without deteriorating the learning effect. However, when it comes to Humanoid and Reacher, it is obvious that there is a great decline in the learning effect of the policy. It mainly owes to the fact that the Policy-wise Diffusion will not diffuse a policy to a certain simple distribution like standard Gaussian, which leads to the policy sampled as $\pi^T$ may not match the true distribution of the diffused policy, therefore causing the failure of the outcome of the naive reverse process. In comparison, the traditional DM, like DDPM has a definite prior distribution in the forward diffusion process, so the fitting of generated samples can be guaranteed.

### A.3.3 OBSERVATIONS ON THE CONDITIONED Q-FUNCTION

In this part, we demonstrate empirical observations on conditioned Q-function to validate Proposition 3.3 and Proposition 3.4. Specifically, we compute average $Q_{\theta,t'}(\pi^\beta|\tilde{\pi})$ for all demonstrations in the dataset, which are viewed as samples collected by all behavior policies $\pi^\beta$, i.e., the policy that is not diffused $\pi^0$. Different levels of noisy demonstrations are sampled as ones collected by its corresponding diffused policy $\tilde{\pi}$ for evaluation. The x-axis in Figure 5 represents different $t'$, and the y-axis is the value of $Q_{\theta,t'}$. The red point is the largest value induced among all $t'$.

As the first row shown in Figure 5, when $t' = t$, it induces the largest value of $Q_{\theta,t'}$, which corresponds to the conclusion of Proposition 3.3. And the second row tells us that 0 makes the largest

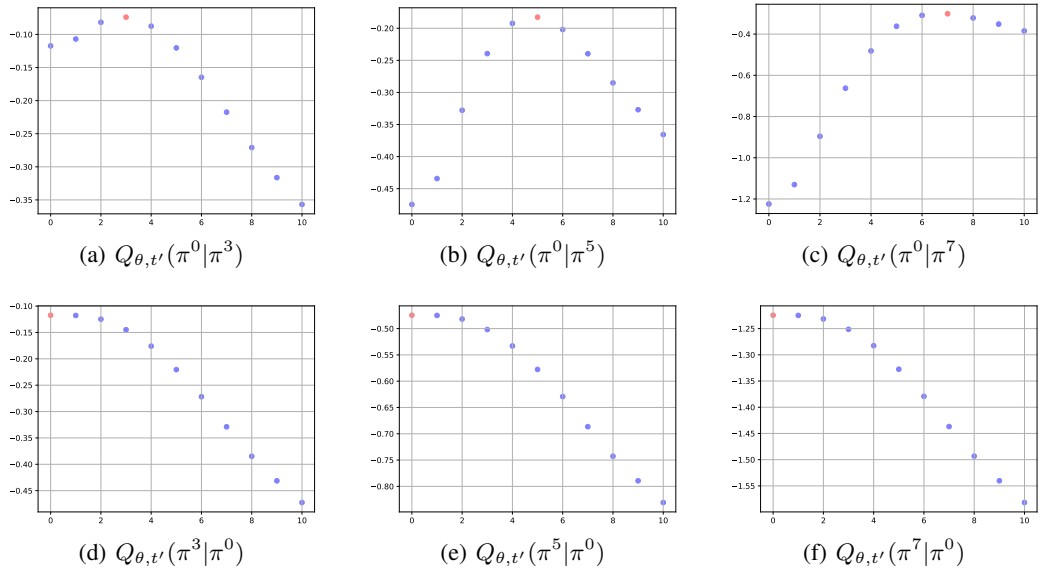

Figure 5: Observations on the value of conditioned Q-function for Hopper

value when $\tilde{\pi}$ is inherently "cleaner" than $\pi^\beta$, which validates Proposition 3.4. Besides, we notice that the value of conditioned Q-function decreases as $t'$ goes away from $t$, which is consistent with our conclusion in the Proof A.1.6.

### A.3.4 RESULTS ON THE EXPERIMENT SETUP OF CHECKPOINTS

In this part, we further evaluate SMILE and other methods with demonstrations collected by checkpoints from different stages during the training of expert policy in order to validate the applicability in practical imitation learning settings.

Fig 6 shows the performance of methods on several MuJoCo tasks. It is obvious that SMILE is still able to learn the expert policy, which further illustrates that SMILE has the potential to capture the unknown complex source that occurred in real-world tasks and resulted in corrupt actions. Besides, it also outperforms other methods on most tasks, validating its effectiveness in more general settings. However, other methods exhibit some performance deterioration on these tasks, suggesting they are relatively vulnerable to noisy demonstrations.

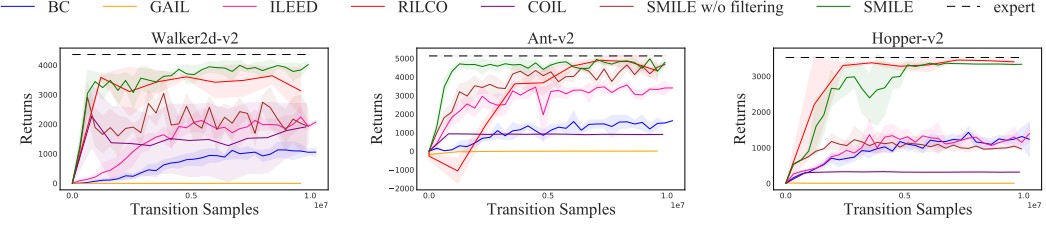

Figure 6: Learning Curves on the checkpoints setup

### A.3.5 PERFORMANCE OF SMILE AND RILCO FOR LESS PROPORTION OF EXPERT DATA

As discussed in Section 4.2, RILCO relies on the assumption that the expert date is more than half among all data. And the unsatisfaction of this on several tasks influences its learning. Regarding this, we also validated our conjecture of the RILCO's underperformance by exploring the impact of expert data proportion on the experimental outcomes of both SMILE and RILCO.

To do so, we conducted evaluations on SMILE and RILCO separately using a demonstration dataset containing less than half of expert demonstrations (5% and 14%). As shown in Fig. 7, our experiments revealed that as the proportion of expert demonstrations decreased, both algorithms exhibited a certain degree of instability and decreased performance. In comparison, SMILE demonstrated relatively better stability, reinforcing the reliability of our conjecture and illustrating better robustness compared to RILCO.

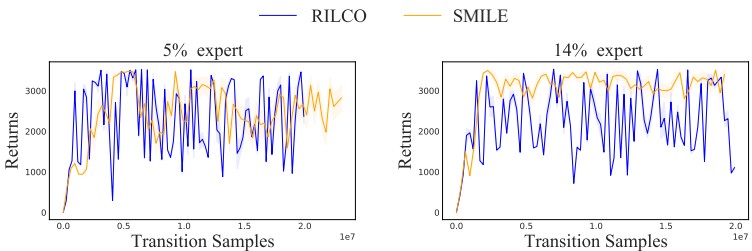

Figure 7: Learning Curves on the different proportions of expert data

### A.3.6 DETAILED EXPLANATION FOR ONE-STEP GENERATOR

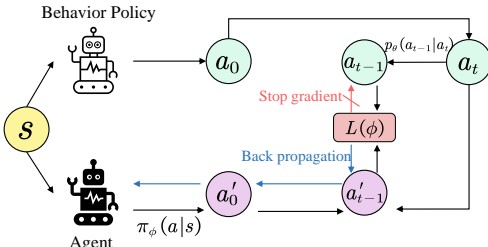

Figure 8: Action $a_0$ will be diffused to $a_t$ according to $q(a_t|a_0, s)$, then $p_\theta$ will be conducted to denoise $a_t$ to $a_{t-1}$. Simultaneously, one-step generator $\pi_\phi$ will generate action $a'_0$ conditioned on $s$. $a'_{t-1}$ is sampled from $q(a'_{t-1}|a_t, a'_0)$ to compare with $a_{t-1}$ and back propagate the gradient to update $\pi_\phi$; stop gradient is needed for avoiding potential issues arising from both models being trained simultaneously.

