# OpenReview forum: "Good Better Best: Self-Motivated Imitation Learning For Noisy Demonstrations"
_ICLR.cc/2024/Conference — Submitted to ICLR 2024_

### Official Review · Reviewer_itCd · 2023-10-30

**Soundness:** 3 good
**Presentation:** 2 fair
**Contribution:** 3 good
**Rating:** 5
**Confidence:** 4

**Summary:**

The paper considers the problem of imitation learning from noisy demonstrations. To address the problem, the paper introduces Self-Motivated Imitation Learning (SMILE), which filters out noisy demonstrations using a diffusion model. Theoretical results are introduced to show the efficacy of the proposed algorithm, and experiments are done to show that SMILE researches higher rewards compared with other baselines.

**Strengths:**

The paper considers an important problem. The proposed algorithm that uses diffusion model to judge the optimality of the demonstrations is novel and interesting. The experimental results show that the algorithm is promising.

**Weaknesses:**

1. The technical writing of the paper can be improved. There are several places that are not fully clear to me:

- Definition 2.1, I think comparing the expertise of two demonstrations by only comparing their rewards are not sufficient. What if the two demonstrations start from different initial conditions? Also, the environment considered is a stochastic environment, where the reward of two trajectories can be different even if we use the same policy. How does the definition deal with this problem?

- Proposition 3.1, in what sense does the author mean by "non-expert"? Can the authors define "non-expert" mathematically first? In addition, in the proof of Proposition 3.1, only action-wise proof is provided. However, is "non-expert" a property that might be defined over trajectories?

- The notation $t$ is a bit confusing. Sometimes the subscript $t$ represents for simulation time step, while sometimes it represents for the diffusion step.

- I encourage the authors to add more explanations to Figure 1, which currently is confusing to me.

2. There are some places for improvement in the experiments:

- It is claimed in the paragraph before "Contributions" that "SMILE achieves results comparable to method that rely on human annotations for several tasks". Which baseline does the authors mean here?

- As introduced in paragraph "Dataset", the experts' original actions are corrupted by adding Gaussian noise, which is consistent to the diffusion model. I wonder what if we corrupt the dataset using other methods? For example, with probability $p$, the agent choose random action.

3. There is some incorrectness and insufficiency of the related work. For example, in the last paragraph of page 1, [1] is introduced as "introduced human annotations to indicate the expertise of the demonstrations" However, I think there is no human annotation in this work, but the algorithm automatically generates labels by injecting noise in the demonstrations itself. Similar works including [2-4] are not included in the related work.

[1] Brown, Daniel S., Wonjoon Goo, and Scott Niekum. "Better-than-demonstrator imitation learning via automatically-ranked demonstrations." Conference on robot learning. PMLR, 2020.

[2] Chen, Letian, Rohan Paleja, and Matthew Gombolay. "Learning from suboptimal demonstration via self-supervised reward regression." Conference on robot learning. PMLR, 2021.

[3] Zhang, Songyuan, et al. "Confidence-aware imitation learning from demonstrations with varying optimality." Advances in Neural Information Processing Systems 34 (2021): 12340-12350.

[4] Xu, Haoran, et al. "Discriminator-weighted offline imitation learning from suboptimal demonstrations." International Conference on Machine Learning. PMLR, 2022.

**Questions:**

Please refer to each point raised in "Weaknesses".

---

> ### Author Response · Authors · 2023-11-17
> **response to Reviewer itCd Pt.1**
>
> We thank the reviewer for the detailed and constructive feedback. Following comments from all reviewers, we revised our manuscript and submitted it. It would be appreciated that if the reviewer could read the revised manuscript.
>
> >**Weakness 1.1**: Definition 2.1, I think comparing the expertise of two demonstrations by only comparing their rewards are not sufficient. What if the two demonstrations start from different initial conditions? Also, the environment considered is a stochastic environment, where the reward of two trajectories can be different even if we use the same policy. How does the definition deal with this problem?
>
> The inclusion of Definition 2.1 serves the primary purpose of elucidating our definition of demonstration expertise, aiming to eliminate any conceptual ambiguity, especially for readers less acquainted with this field. Evidently, the reward signal provided by the environment stands as the most objective and approriate metric for evaluating the demonstrator's expertise explicitly considering they are positively related. Furthermore, while returns under the same policy may exhibit slight variations, the overall differences are negligible, indicating a comparable level of implicit policy expertise. In summary, we find it reasonable to employ the return as an assessment metric for expertise.
>
> >**Weakness 1.2**: Proposition 3.1, in what sense does the author mean by "non-expert"? Can the authors define "non-expert" mathematically first? In addition, in the proof of Proposition 3.1, only action-wise proof is provided. However, is "non-expert" a property that might be defined over trajectories?
>
> As mentioned in the first paragraph of the introduction, an expert represents the optimal behavior policy among all demonstrators. In contrast, all other policies are defined as non-expert policies. We sincerely appreciate your meticulous checking of the proof for Proposition 3.1. As you pointed out, the action-wise proof was not exhaustive. We have fixed this proof by illustrating it in policy-wise level, as indicated by the modifications highlighted in blue within the appendix.
>
> >**Weakness 1.3**: The notation t is a bit confusing. Sometimes the subscript t represents for simulation time step, while sometimes it represents for the diffusion step.
>
> Upon careful proofreading, there are two instances where the term "simulation time step" is mentioned. In the second paragraph under the "notation" section, $n$ is used to represent the simulation time step. The other instance is in the last paragraph of section 3.3, where an superscript $(i)$ is used. Therefore, in this paper, $t$ should exclusively refer to the diffusion step. If there are any other unnoticed instances of wrong usage, please notify us, and we will promptly rectify them in the paper.
>
> >**Weakness 1.4**: I encourage the authors to add more explanations to Figure 1, which currently is confusing to me.
>
> We have augmented the comprehensive description of the overall workflow in Figure 1 and have also made corresponding updates in the appendix section of the paper. These modifications are clearly indicated using green font. We kindly invite you to review these changes.

---

> ### Author Response · Authors · 2023-11-17
> **response to Reviewer itCd Pt.2**
>
> >**Weakness 2.1**: It is claimed in the paragraph before "Contributions" that "SMILE achieves results comparable to method that rely on human annotations for several tasks". Which baseline does the authors mean here?
>
> The baseline we referred to is COIL, which utilizes the reward signal provided by the environment as additional annotation. This aspect has been extensively elaborated as mentioned in the "baseline" of Section 4.1 in this paper.
>
> >**Weakness 2.2**: As introduced in paragraph "Dataset", the experts' original actions are corrupted by adding Gaussian noise, which is consistent to the diffusion model. I wonder what if we corrupt the dataset using other methods? For example, with probability p, the agent choose random action.
>
> Firstly, though noisy demonstraions are collected by adding Gaussian noise, the intention that Gaussian is used there is different from diffusion model. **Specifically, in policy-wise diffusion module, Gaussian is used to model the source of corruption which is subject to an unknown complex distribution. However, during the data collection phase, Gaussian is solely utilized to gather multi-expertise demonstrations**. It's important to note that we do not assess the diffusion step between these collected demonstrations; instead, our focus lies in assessing the diffusion step between the collected demonstrations and $\pi_\phi$ . And there exists no Gaussian linkage between the collected demonstrations and $\pi_\phi$ .
>
> Indeed, there are various ways to corrupt datasets, and we are also curious about how SMILE performs on other noisy demonstrations. Therefore, **we opted to assess our approach using different checkpoints saved during the policy training process, which provide noisy demonstrations with more complex sources of corruption, as demonstrated in our previously provided Appendix A.3.4**. The experiments indicate that, when dealing with more complex noisy demonstrations, SMILE consistently demonstrates considerable performance and still show the ability to discern the expertise of demonstrations.
>
> >**Weakness 3**.There is some incorrectness and insufficiency of the related work. For example, in the last paragraph of page 1, [1] is introduced as "introduced human annotations to indicate the expertise of the demonstrations" However, I think there is no human annotation in this work, but the algorithm automatically generates labels by injecting noise in the demonstrations itself. Similar works including [2-4] are not included in the related work.
>
> We sincerely appreciate your identification of this error, and we apologize for the citation mistake. We will relocate the reference to DREX[1] to the "related work" section and provide additional information about the mentioned SSRR[2]. Due to space constraints, we regret that we cannot include all these literature. All updates addressing this issue will be highlighted in blue font.
>
> [1] Brown, Daniel S., Wonjoon Goo, and Scott Niekum. "Better-than-demonstrator imitation learning via automatically-ranked demonstrations." Conference on robot learning. PMLR, 2020.
>
> [2] Chen, Letian, Rohan Paleja, and Matthew Gombolay. "Learning from suboptimal demonstration via self-supervised reward regression." Conference on robot learning. PMLR, 2021.
>
>
> Again, thank you for your valuable comments. Please do let us know if you have any remaining questions.
>
> Best, Authors

---

> > ### Comment · Reviewer_itCd · 2023-11-22
> > **Response to authors**
> >
> > I want to thank the authors for their reply. The reply has addressed some of my concerns, but I still have the following comments:
> >
> > Weakness 1.1: In the literature, it is common to define the expertise based on the policy, i.e., if one policy has a higher expected reward, it is better than the other. In the MDP setting, I believe it is essential to compare the expectations, instead of the reward of a single trajectory. If we use the current definition, one policy can have higher/lower expertise than itself, depending on different initial states and random seeds. I think this is not proper.
> >
> > Weakness 1.4: It might be better if the authors directly add the explanations of Figure 1 in the main texts.
> >
> > Weakness 2.1: It is true that in some cases "reward signal provided by the environment" is a kind of "human annotation", however, "human annotations" are more related to preference-based learning in literature, or baselines that really use human annotations, for example, T-REX. I encourage the authors to compare with these baselines, or refine the word "human annotation".
> >
> > Best,
> > Reviewer itCd

---

> > > ### Author Response · Authors · 2023-11-22
> > > **Response to reviewer**
> > >
> > > Thank you for your prompt response and efforts. Your insights have significantly contributed to enhancing the quality of our work. We found your latest points quite valuable, and we've promptly updated our manuscript accordingly. Here are our responses to the specific details:
> > >
> > > **Weakness 1.1**: We greatly appreciate your meticulousness and thoroughness on defining policy expertise. We agree that your proposed definition helps clarify conceptual ambiguities. Consequently, we've made modifications in the main text to align with this definition.
> > >
> > > **Weakness 1.4**: We acknowledge the preference to move the explanation of Figure 1 to the main text. However, due to the stringent page limitations imposed by ICLR, restricting the main text to a maximum of nine pages across all versions, we reluctantly made this compromise.
> > >
> > > **Weakness 2.1**:  We're grateful for you highlighting the ambiguity in describing COIL using human annotation. Consequently, we have made the necessary revisions in the manuscript to rectify this.
> > >
> > > All the updates have been highlighted in blue font. Should you still have any confusion or further suggestions, please feel free to express them. We assure you that we will earnestly consider any additional feedback you provide.
> > >
> > > Besr, authors

---

### Official Review · Reviewer_z47T · 2023-10-31

**Soundness:** 3 good
**Presentation:** 3 good
**Contribution:** 3 good
**Rating:** 6
**Confidence:** 3

**Summary:**

This paper uses diffusion model in place of GAN in generative adversarial imitation learning problem. At the first stage, this paper uses diffusion model to learn the noise information for forward and reverse process on the expert demo. Then, the noise information is leveraged to predict the diffusion steps between the current policy and demonstrators. Experiments show that this work have some performance gain s upon noisy expert demonstrations.

**Strengths:**

This work is novel and easy to follow. I think diffusion model is applied here do have some advantages. For example, imitation learning could be more robust to the noisy expert demos.

**Weaknesses:**

1. The performance gains seem little.
2. I think there are some methods focusing on noisy expert imitation. Have the authors surveyed these methods and do a comparison?
3. I would like to see the experiment results based on clean expert data. I am wondering wether diffusion model has some advantages compared to generative models in imitaiton learning when the expert data is clean.

**Questions:**

Could the authors report the training time of this newly proposed method. I think diffusion model is too slow for training in imitation learning setting. I am concerned about this. However, I would like to see more results with clean and noisy expert demos in experiments. I am wondering why diffusion model could be better than generative model such as GAIL, except for noisy expert setting. Could the author illustrate this?

---

> ### Author Response · Authors · 2023-11-17
> **response to Reviewer z47T**
>
> We thank the reviewer for the detailed and constructive feedback. Following comments from all reviewers, we revised our manuscript and submitted it. It would be appreciated that if the reviewer could read the revised manuscript.
>
> >**Weakness 1**.The performance gains seem little.
>
> While SMILE may gain "little" advantage than other methods on all tasks, **this does not diminish the strength of our approach**. The key advantage of SMILE is its ability to automatically filter out noisy demonstrations and learn expert policies without relying on annotations, rather than simply optimizing performance based on any other comparison algorithms. On the contrary, our intention is to offer a valuable solution that promotes the refinement of techniques without relying on additional annotation. In Section 4.2, we demonstrate SMILE's ability to learn expert policies across a spectrum of demonstrations expertise through empirical evaluation. Furthermore, Section 4.3 provides evidence of SMILE's ability to effectively filter out noisy demonstrations during training.
>
> >**Weakness 2**.I think there are some methods focusing on noisy expert imitation. Have the authors surveyed these methods and do a comparison?
>
> We have extensively researched the field of noisy imitation learning, carefully selected algorithms that, like ours, handle noisy demonstrations without annotations for comparison. In the related work section, we have further engaged in a detailed discussion of methods, whether annotation is used or not. Additionally, we have introduced several representative solutions employing diffusion models in imitation learning. Admittedly, our knowledge is limited, and due to page limitations, some methods with less relationship to ours may not have been explicitly mentioned. If there are noteworthy works worth mentioning, please feel free to suggest them, and we will incorporate them into the main text.
>
> >**Weakness 3**.I would like to see the experiment results based on clean expert data. I am wondering wether diffusion model has some advantages compared to generative models in imitaiton learning when the expert data is clean.
>
> The potential of the diffusion model, as a generative model, in the context of Imitation Learning (IL) is evident. As introduced in the “preliminary” of SMILE, the diffusion model exhibits a commendable capability to fit the distribution of policies[1]. Given that IL heavily relies on offline datasets, and the policy distribution tends to be intricate, the diffusion model should perform exceptionally well in scenarios where demonstrators are clean. Considering the existing body of work, and our specific focus on addressing noisy demonstrations, we acknowledge that the performance of the diffusion model on clean data is not the primary emphasis of our study. Therefore, we have chosen to keep our focus on the IL in noisy demonstrations setting.
>
> >**Question 1**: Could the authors report the training time of this newly proposed method. I think diffusion model is too slow for training in imitation learning setting. I am concerned about this. However, I would like to see more results with clean and noisy expert demos in experiments. I am wondering why diffusion model could be better than generative model such as GAIL, except for noisy expert setting. Could the author illustrate this?
>
> In our current hardware setup (RTX 3090) and with the specified parameter configurations, the overall training time for SMILE ranges from 6 to 10 hours. This duration is primarily contingent on the frequency of model evaluation and the occupancy of hardwares, which corresponds to the frequency at which the policy interacts with the environment. Furthermore, with respect to the comparative analysis of the performance of generative models based on clean data, as mentioned in our response to weakness 3, there are existing studies [1, 2] that delve into such issues. Hence, we have chosen not to include further details regarding this matter within our paper.
>
> [1] Pearce T, Rashid T, Kanervisto A, et al. Imitating human behaviour with diffusion models[J]. arXiv preprint arXiv:2301.10677, 2023.
>
> [2] Wang Z, Hunt J J, Zhou M. Diffusion policies as an expressive policy class for offline reinforcement learning[J]. arXiv preprint arXiv:2208.06193, 2022.
>
> Again, thank you for your valuable comments. Please do let us know if you have any remaining questions.
>
> Best, Authors

---

> > ### Comment · Reviewer_z47T · 2023-11-23
> > **Thanks for your response**
> >
> > Thanks for your response. I would like to keep my score.

---

### Official Review · Reviewer_76Xe · 2023-11-03

**Soundness:** 2 fair
**Presentation:** 2 fair
**Contribution:** 2 fair
**Rating:** 3
**Confidence:** 5

**Summary:**

The paper introduces a novel method called Self-Motivated Imitation Learning (SMILE) for imitation learning in situations where there are varying levels of expertise in the demonstrations provided. The main contribution is the ability of SMILE to predict the number of diffusion steps (akin to the level of noise) between the current policy and the demonstrations, which correlates to the expertise gap. The authors theoretically justify their approach and provide a detailed explanation of how this prediction mechanism works for filtering purposes. They then validate their method through experiments on MuJoCo tasks. The results show that SMILE can effectively learn from the best available demonstrations and ignore those that are less skilled, leading to more efficient learning of expert policies.

**Strengths:**

- provide the proof of predicting how many steps to denoise
- the results is better when the non-expert is just expert plus noise generated by Gaussian distribution

**Weaknesses:**

- The paper claims that they want to handle non-expert demonstrations. However, the non-expert demonstrations they handle are only demonstrations generated by the same expert but some gaussian noise. There are many other ways to generate non-expert trajectories.For example, one can perturb the input observation and get a perturbed action. In addition, dataset D4RL provides non-expert demonstrations directly.  Many other methods have shown the ability to handle those non-expert demonstrations.
- There can be multiple kinds of experts in Mujoco. The proposed method might learn only one of them and be unable to handle the states of other experts.
- Since the method filters out many demonstrations, it might lose the chance to learn the dynamic of the environment and ends up being bad at OOD states.
- The many parts of the design are different from DDPM. The author needs to provide explanations. For example, in eq.6, q(a_t|a_{t-1}, s) is different from ddpm (eq.3). Another example is that it uses a one-step generator. I wonder about the performance of it compared to multisteps. Especially if it uses DDIM.

**Questions:**

- It is hard to understand the one-step generator. What is \mu_t in equation 10? Why not just train an additional policy with algorithms like BC and the data that have been filtered.

---

> ### Author Response · Authors · 2023-11-17
> **response to Reviewer 76Xe Pt.1**
>
> We thank the reviewer for the detailed and constructive feedback. Following comments from all reviewers, we revised our manuscript and submitted it. It would be appreciated that if the reviewer could read the revised manuscript.
>
> > **Weakness 1**: The paper claims that they want to handle non-expert demonstrations. However, the non-expert demonstrations they handle are only demonstrations generated by the same expert but some gaussian noise. There are many other ways to generate non-expert trajectories.For example, one can perturb the input observation and get a perturbed action. In addition, dataset D4RL provides non-expert demonstrations directly. Many other methods have shown the ability to handle those non-expert demonstrations.
>
> Indeed, there are various ways to corrupt datasets, and we are also curious about how SMILE performs on other noisy demonstrations. **Therefore, we opted to assess our approach using different checkpoints saved during the policy training process, which provide noisy demonstrations with more complex sources of corruption, as demonstrated in the previous Appendix A.3.4**. The experiments indicate that, when dealing with more complex noisy demonstrations, SMILE consistently demonstrates considerable performance. Besides, we acknowledge that there are alternative approaches to addressing noisy demonstrations without relying on annotations. And we have extensively cited and compared methods that we know, including a comparative analysis with representative work like RILCO and ILEED. We are thankful if you would share any other noteworthy work, and we will certainly include them in our paper.
>
> >**Weakness 2**: There can be multiple kinds of experts in Mujoco. The proposed method might learn only one of them and be unable to handle the states of other experts.
>
> It is important to emphasize that our theoretical framework does not assume the uniqueness of the expert policy. In other words, any form of an expert policy is applicable within our theoretical framework. Therefore, as long as SMILE accurately distinguish expert and non-expert demonstrations, it retains the expert demonstrations in the dataset for learning, regardless of the number of the kinds of experts. Our experiments have demonstrated SMILE's remarkable ability to capture the expertise within demonstrations, as detailed in section 4.3. Moreover, irrespective of the diversity of experts, as long as their states are present in the dataset, SMILE effectively learns the corresponding expert decisions in those states. Furthermore, our experiments in section 4.2 showcase SMILE's superior stability during evaluation across most environments compared to other algorithms. This suggests that even when encountering unseen states in training dataset during evaluation, SMILE generally handles them well.
>
> >**Weakness 3**: Since the method filters out many demonstrations, it might lose the chance to learn the dynamic of the environment and ends up being bad at OOD states.
>
> Firstly, as our demonstrations are collected by corrupted policies during environmental interaction, the trajectory collection process still subject to environmental dynamics. Furthermore, before the filtering happens, there is still ample opportunity to learn the dynamics. It's important to point out that our filtering of noisy demonstrations in the dataset aims to prevent their influence on policy learning, but it doesn't imply that we can't continue utilizing them to assist in learning about dynamics. Therefore, there is no missed opportunity to learn from it. Additionally, learning environmental dynamics in an offline setting and handling out-of-distribution states are independent and substantial topics even if the number of demonstrations is abundant. Given that this paper primarily focuses on safeguarding policies trained without being influenced by noisy demonstrations, we do not intend to delve extensively into these aspects at this stage. Of course, we acknowledge that these remain pressing issues in imitation learning, and in our future work, we will continue to improve our algorithm to effectively learning the environment dynamic as well.

---

> ### Author Response · Authors · 2023-11-17
> **response to Reviewer 76Xe Pt.2**
>
> > **Weakness 4**: The many parts of the design are different from DDPM. The author needs to provide explanations. For example, in eq.6, q(a_t|a_{t-1}, s) is different from ddpm (eq.3). Another example is that it uses a one-step generator. I wonder about the performance of it compared to multisteps. Especially if it uses DDIM.
>
> The framework design of SMILE differs from DDPM because, although they share many similarities, they are different work. **This distinction primarily arises because the diffusion process in DDPM is geared towards diffusing samples into a specific prior distribution, whereas the policy-wise diffusion process in SMILE is aimed at simulating the corruption of policy expertise**. This fundamental difference in purpose leads to variations in the framework design. Regarding the ablation study on whether SMILE employs a one-step generator, we believe Appendix 3.2 have already provided detailed empirical results and corresponding analyses.
>
> >**Question 1**: It is hard to understand the one-step generator. What is \mu_t in equation 10? Why not just train an additional policy with algorithms like BC and the data that have been filtered.
>
> As mentioned in the paragraph containing Eq. (10), $mu_t$ represents the mean of the two distributions in Eq. (9). We directly cited this for computational convenience, similar to its use in DDPM.
>
> Regarding the significance of the one-step generator, the reverse process of the diffusion model enhances the expressiveness of the generator remarkably compared to other generative models such as GAIL, as discussed in DDPM and DBC[1]. We believe that the diffusion model has better potential to fit the complex policy distribution from the offline demonstrations which might be complex to learn. However, since the generation process of the diffusion model is multi-step, this inevitably leads to a decrease in the decision-making efficiency of the agent during interaction. Hence, we devised the one-step generator to encourage it to predict the outcomes of the multi-step generator, ensuring model expressiveness without incurring a substantial increase in time cost.
>
> [1] Pearce T, Rashid T, Kanervisto A, et al. Imitating human behaviour with diffusion models[J]. arXiv preprint arXiv:2301.10677, 2023.
>
> Again, thank you for your valuable comments. Please do let us know if you have any remaining questions.
>
> Best, Authors

---

> ### Comment · Reviewer_76Xe · 2023-12-01
> **comments**
>
> W1: using the checkpoint of a same model is not diverse enough. [1] shows that an agent can be bad other agent's states even if they are train with the same algorithm.
>
> W2: How diverse are you experts?
>
> W3: The agent might forget during training.
>
> [1] Can Agents Run Relay Race with Strangers? Generalization of RL to Out-of-Distribution Trajectories, ICLR 2023

---

### Official Review · Reviewer_Qmx6 · 2023-11-10

**Soundness:** 2 fair
**Presentation:** 2 fair
**Contribution:** 2 fair
**Rating:** 3
**Confidence:** 4

**Summary:**

In this paper, the authors address the challenge of noisy demonstrations in Imitation Learning (IL), which hinders the discovery of effective policies. They propose Self-Motivated Imitation Learning (SMILE), a method that progressively filters out demonstrations from policies considered inferior to the current policy, eliminating the need for additional information about the demonstrators' expertise. SMILE leverages Diffusion Models to simulate the shift in demonstration expertise, extracting noise information that diffuses expertise from low to high and vice versa. The predicted diffusion steps are used to filter out noisy demonstrations in a self-motivated manner, as empirically demonstrated on MuJoCo tasks, showing proficiency in learning expert policies amidst noisy demonstrations.

**Strengths:**

* This paper employs a diffusion model, which has shown promising performance in generative model training. The idea of this paper seems to be novel.

* The authors provide theoretical derivations and empirical results demonstrate good results.

**Weaknesses:**

1. The paper's clarity can be significantly enhanced. For instance, the caption of Figure 1 lacks sufficient information for readers to fully comprehend its content. Furthermore, there is a need for a detailed explanation how does SMILE algorithm actually perform and how the noisy demonstrations filter is incorporated into existing IL methods. The methodology section lacks an overall algorithmic explanation, causing confusion. While the appendix provides pseudocode to elucidate the algorithm, the authors should emphasize these details in the main methodology section. Additionally, the authors introduce Definition 2.1 in the preliminary part, but its application in the subsequent content remains unclear.

2. The authors should provide more details about the dataset used for training since they collect the dataset themselves. For example, the quality of each inferior demonstrator related to varying levels, the number of demonstrations used for training should be provided. Moreover, does the corrupted action being used to transit to the new state when collection demonstration?

3. My critical concern is about the way the suboptimal data is generated. The method is to add Gaussian noise to the actions of an optimal policy. This noise maps exactly the one used in the diffusion process. Is this a relevant factor to explain the performance of the method? It would be great to investigate other forms of noise.

4. The evaluations are only conducted on MuJoCo tasks. Is it able to evaluate the proposed method using one of the many existing datasets of human demos, such as RoboMimic? RoboMimic includes a classification of the level of dexterity of human demonstrations in multiple robotic tasks (in simulation), akin to the levels of noise used in the paper's experiments. Are there additional issues or limitations when applying this method to human-generated data?

**Questions:**

1. From the pseudecode provided in the appendix, it seems that SMILE can be incorporated with both GAIL and BC. However, it's unclear which IL method is used to incorporate with SMILE in Figure 2. If BC is employed, it might introduce a potential fairness issue when comparing it with GAIL. Additionally, is it possible to integrate the SMILE method with online methods, and if so, what could be the expected performance?

2. I believe VILD [1] in online setting or modified VILD in offline setting (using pre-collected demonstrations and using IQL or CQL instead of SAC or TRPO) can serve as a powerful baseline, both theoretically and experimentally.

3. I am wondering if it is suitable to connect the proposed method to the idea of self-paced learning. Self-paced learning starts from easier sample (which is judged by the sample loss) and gradually include more samples into training to ensure the generalization. In SMILE, the authors seem to start from the whole dataset and gradually filter out noisy demonstrations.

4. According to Algorithm 2, while both diffusion model and policy network are initialised, how could the algorithm achieve good performance at filtering out noisy demonstrations? Additionally, is there any theoretical guarantee for the convergence of the diffused policy and the agent policy?

[1] Variational Imitation Learning with Diverse-quality Demonstrations, ICML 2020.

[2] DemoDICE: Offline Imitation Learning with Supplementary Imperfect Demonstrations, ICLR 2022.

---

> ### Author Response · Authors · 2023-11-17
> **response to Reviewer Qmx6 Pt.1**
>
> We thank the reviewer for the detailed and constructive feedback. Following comments from all reviewers, we revised our manuscript and submitted it. It would be appreciated that if the reviewer could read the revised manuscript.
>
> >**Weakness 1**.The paper's clarity can be significantly enhanced. For instance, the caption of Figure 1 lacks sufficient information for readers to fully comprehend its content. Furthermore, there is a need for a detailed explanation how does SMILE algorithm actually perform and how the noisy demonstrations filter is incorporated into existing IL methods. The methodology section lacks an overall algorithmic explanation, causing confusion. While the appendix provides pseudocode to elucidate the algorithm, the authors should emphasize these details in the main methodology section. Additionally, the authors introduce Definition 2.1 in the preliminary part, but its application in the subsequent content remains unclear.
>
> Thank you for highlighting the lack of clarity in Figure 1. We've addressed this concern by providing additional explanations in the text for Figure 1. Due to space constraints, we placed these details in Appendix A.3.6, colored in green for your reference. Additionally, to better explain how SMILE operates, we added brief explanations in the main text, referencing the pseudocode's location and providing further details in the corresponding location, all highlighted in red.
>
> Furthermore, there might be some misunderstanding with the generation process of SMILE. Although BC, GAIL, and SMILE all have their policy trained to generate actions for decision making, **SMILE, in essence,  holds an independent and unique generation approach compared to BC and GAIL**. Specifically, we opted to follow the generation method employed by DDPM so that our trained policy could better fit the complex policy distribution through noisy demonstrations. In implementation, we additionally devised a one-step generator to modify original generation process of diffusion model for improved decision-making efficiency. This fundamentally sets SMILE apart from other IL methods like GAIL.
>
> We acknowledge that we did not explicitly refer to Definition 2.1 in the paper. Its inclusion aims to clarify our definition of demonstration expertise to avoid any conceptual ambiguity for readers less familiar with this field.
>
> >**Weakness 2**.The authors should provide more details about the dataset used for training since they collect the dataset themselves. For example, the quality of each inferior demonstrator related to varying levels, the number of demonstrations used for training should be provided. Moreover, does the corrupted action being used to transit to the new state when collection demonstration?
>
> We only briefly discussed the quality of inferior demonstrators as mentioned in "datasets" in section 4.1 due to page limits. In detail, Gaussian noise that is adopted  to corrupt the dataset is at varying levels range linearly increased from 0 to 1. And there are 100 multi-expertise demonstrations in total. We have taken into account your suggestion and provided additional details in the main text, which are highlighted in red. Specifically, we introduce perturbations **during the data collection**, which means we add noise to expert action to obtain non-expert action, and then submit it to the environment to obtain the next observation. Therefore, the corrupted action will naturally be used to transit to the new state.
>
> Furthermore, regarding the number of demonstrations used during the training process, we have provided detailed explanations in the "datasets" section in Section 4.1: "We collected ten trajectories for each noise level."

---

> > ### Comment · Reviewer_Qmx6 · 2023-11-23
> > **Thanks for the rebuttal**
> >
> > I appreciate the detailed feedback from the reviewer. However, I still have some questions and would appreciate further clarification from the authors.
> >
> > W1: The caption of Figure 8 is still confusing to me, particularly in relation to the sampling of $a^\prime_{t−1}$ from $q(a^\prime_{t−1}|a_t, a^\prime_0)$. I was unable to locate the definition of this $q$ in the paper (I briefly check the paper this time and might overlook some details). Could the authors provide a more explicit explanation of how $a^\prime_{t−1}$ is derived from $a_t$? Additionally, I would like clarification on how the ground-truth denoiser $p_\theta$ is obtained. Is it trained on an extra clean datasets?
> >
> > W2: Normally we refer 1 demonstration as 1 state-action pair. As the authors claim that "We collected ten trajectories for each noise level, ultimately building a complete dataset containing 100 demonstrations in total", does it mean the authors use 100 expert trajectories for training? Considering the each expert trajectories can contain up to 1000 demonstrations, this seems to make the whole dataset quite big.

---

> > > ### Author Response · Authors · 2023-11-23
> > > **Response to reviewer**
> > >
> > > We are more than willing to address these queries:
> > >
> > > **W1**: Regarding the expression of $q(a_{t-1}|a_t,a_0)$, we indeed did not explicitly provide its specific form. This omission primarily arose due to page constraints. In our manuscript, we opted to reference the explanation given in the original DDPM (see the paragraph above Eq,9). Specifically, $q(a_{t-1}^\prime|a_t,a_0^\prime)$ approximates the ground-truth denoising process of the diffusion model, denoted as $p(a_{t-1}|a_{t})$. It is derived from the Bayesian:
> > >
> > > $q(a_{t-1}|a_t,a_0)=q(a_t|a_{t-1},a_0)\frac{q(a_{t-1}|a_0)}{q(a_t|a_0)}$.
> > >
> > > It's observed that it involves multiple Gaussian operations, thus making $q(a_{t-1}|a_t,a_0)$ also Gaussian. To simplify the derivation process and adhere to DDPM's description, we denote it as $q(a_{t-1}|a_t,a_0)=\mathcal{N}(a_{t-1};\mu_t(a_t,a_0),\tilde{\beta}_t \mathbf{I})$. Furthermore, we outline the specific form of $\mu_t(a_t,a_0)$ for its utilization in Eq. 10:
> > >
> > > $\mu_t(a_t,a_0)=\frac{\alpha_{t-1}^2}{\sigma_t^2}x_t-\frac{\beta_t^2}{\sigma_t^2}x_0$.
> > >
> > > Regarding the denoiser $p_\theta$, it's essentially induced by the noise approximator $\epsilon_\theta$. Additionally, as explained in our response to Question 4 and as illustrated in the pseudocode in Algorithm 1, $\epsilon_\theta$ is not trained on a separate clean dataset. Instead, both $\epsilon_\theta$ and $\pi_\phi$ are trained on exactly the same noisy dataset simultaneously. In the initial stages of training, it's evident that $p_\theta$ has not yet reached the 'ground-truth'. However, we aim for it to guide $\pi_\phi$ during these initial stages. Therefore, we opted for $\epsilon_\theta$ to update n times more than $\pi_\phi$ at each iteration during training (n being an adjustable hyper-parameter). The specific details behind this approach are elaborated in our response to Question 4.
> > >
> > > **W2**: Firstly, within the context of Imitation Learning (IL), 'demonstration' typically refers to a complete interaction trajectory collected by a demonstrator, rather than a single state-action pair. Therefore, our dataset is not extensive in terms of volume. To provide specifics, across 10 levels of demonstrators, there are a total of 10 trajectories collected by each demonstrator, resulting in a combined count of 100 trajectories in the dataset. To the best of our knowledge, such a dataset size is widely acceptable within IL methodologies.
> > >
> > > Should you still have any confusion or further suggestions, please feel free to express them. We assure you that we will earnestly consider any additional feedback you provide.
> > >
> > > Besr, authors

---

> ### Author Response · Authors · 2023-11-17
> **response to Reviewer Qmx6 Pt.2**
>
> >**Weakness 3**.My critical concern is about the way the suboptimal data is generated. The method is to add Gaussian noise to the actions of an optimal policy. This noise maps exactly the one used in the diffusion process. Is this a relevant factor to explain the performance of the method? It would be great to investigate other forms of noise.
>
> While in the "experiments" section, we introduced noisy demonstrations by using Gaussian noise to corrupt the expert policy, we do not consider this method as a relevant factor explaining the performance of our approach. **Specifically, in policy-wise diffusion module, Gaussian is used to model the source of corruption which is subject to an unknown complex distribution. However, during the data collection phase, Gaussian is solely utilized to gather multi-expertise demonstrations**. It's important to note that we do not assess the diffusion step between these collected demonstrations; instead, our focus lies in assessing the diffusion step between the collected demonstrations and $\pi_\phi$ . And there exists no Gaussian linkage between the collected demonstrations and $\pi_\phi$ .
>
> Technically, the noise approximator $\epsilon_\theta$  in the filter module takes the actions $a^{\pi_\phi}$  generated by $\pi_\phi$ as input, and $a^{\pi_\phi}$  is directly generated by the on-training policy, not obtained by adding Gaussian noise to any specific policy. This is distinct from the scenario of training the noise approximator to capture the corruption in policy expertise, where the input of $\epsilon_\theta$  during training are diffused actions (modeled as non-expert actions to simulate the corruption of the expertise).
>
> Besides, we have also explored SMILE's performance on other forms of inferior demonstrations, as mentioned in Appendix A.3.4. When demonstrations in the dataset are collected by multiple checkpoints from initial policy trining stage to the end of training, which provide noisy demonstrations with more complex sources of corruption, SMILE continues to learn expert-level policies and demonstrates superior performance compared to other algorithms.
>
> >**Weakness 4**: The evaluations are only conducted on MuJoCo tasks. Is it able to evaluate the proposed method using one of the many existing datasets of human demos, such as RoboMimic? RoboMimic includes a classification of the level of dexterity of human demonstrations in multiple robotic tasks (in simulation), akin to the levels of noise used in the paper's experiments. Are there additional issues or limitations when applying this method to human-generated data?
>
> Thans for the suggestion. We are currently conducting this experiment, aiming to provide additional results before the deadline. Once we have get the outcomes, we will promptly report them.
>
> >**Question 1**: 1.From the pseudecode provided in the appendix, it seems that SMILE can be incorporated with both GAIL and BC. However, it's unclear which IL method is used to incorporate with SMILE in Figure 2. If BC is employed, it might introduce a potential fairness issue when comparing it with GAIL. Additionally, is it possible to integrate the SMILE method with online methods, and if so, what could be the expected performance?
>
> The special generative and training approach of SMILE determines that it is not suitbale to incorporate existing IL methods (where BC directly uses MSE or MLE for training, and GAIL utilizes discriminator-provided ratings for IRL). SMILE, on the other hand, maintains the original generative process of DDPM and modifies it by employing the noise approximator $\epsilon_\theta$  to guide the training of $\pi_\phi$, predicting samples generated by a multi-step diffusion chain. **In essence, the generation process of SMILE is equivalent to DDPM's and other generative models like GAN train their generator in their own different ways**.
>
> Besides, SMILE is designed as an algorithm that enhances imitation learning robustness without the requirement for annotations, while online setting requires reward signals from the environment or reward functions trained by another auxilary task to train the value network, which conflicts with our purpose of reducing the overhead and may compromise the sample efficiency. Therefore, the main application of SMILE is still on offline setting. However, considering that the metric for filtering trajectories in SMILE involves assessing the diffusion step between demonstrations and $\pi_\phi$, as explained in "observation2" in section 4.3, we have also considered utilizing the diffusion step to train the reward function. Nevertheless, we believe that this idea is not mature enough and is still in the exploratory phase. Additionally, based on our observations of the IRL algorithm RILCO, if policy training is conducted in an online setting, it might lead to faster convergence but could also introduce training instabilities, resulting in fluctuations in the learning curve.

---

> ### Author Response · Authors · 2023-11-17
> **response to Reviewer Qmx6 Pt.3**
>
> >**Question 2**.I believe VILD [1] in online setting or modified VILD in offline setting (using pre-collected demonstrations and using IQL or CQL instead of SAC or TRPO) can serve as a powerful baseline, both theoretically and experimentally.
>
> We acknowledge VILD as a significant inspiration for our work. However, in selecting the baseline, we opted for ILEED over VILD because the two methods are quite similar, yet ILEED is relatively more aligned with SMILE's offline setting. **Rougly, ILEED could be viewed as modified VILD in offline setting**. Specifically, both ILEED and VILD consider leveraging the identities of demonstrators to assess their expertise. However, the divergence lies in the details: ILEED conducts additional state feature extraction on the state to evaluate the "skill proficiency" of the demonstrator across different states. Furthermore, both propose Gaussian modeling of non-expert policy. The primary distinction lies in ILEED's direct prediction of optimal behavior based on the data itself, constituting an offline setting algorithm. Conversely, VILD involves training a reward function using expertise, representing an online setting approach. Last but not least, as our comparison already contained an online setting algorithm (RILCO), we eventually opted for ILEED in this paper.
>
> >**Question 3**.I am wondering if it is suitable to connect the proposed method to the idea of self-paced learning. Self-paced learning starts from easier sample (which is judged by the sample loss) and gradually include more samples into training to ensure the generalization. In SMILE, the authors seem to start from the whole dataset and gradually filter out noisy demonstrations.
>
> Your keen grasp of the distinction between self-paced learning and self-motivated learning is impressive. But at this stage, we still believe that elucidating the connection and differences between the two in our paper is crucial. Clearly, while they entail distinct motivations and implementation methods, there are also conceptual similarities—both involve selecting samples based on a predefined metric and model itself. **To wrap up, we think our method brought a brand new view to self-paced learning, which is about how to distinguish "easy" samples (less-expertise demonstrations) from the "hard" ones (more-expertise demonstrations)  in decision-making situation**. With this concern in mind, we consider clarifying the relationship and disparities between them in our paper vital to preventing any conceptual confusion among readers and ensuring a clear understanding of our paper.
>
>
> >**Question 4**.According to Algorithm 2, while both diffusion model and policy network are initialised, how could the algorithm achieve good performance at filtering out noisy demonstrations? Additionally, is there any theoretical guarantee for the convergence of the diffused policy and the agent policy?
>
> We sincerely appreciate your keen insights. At the outset of algorithm design, we shared similar concerns about the accuracy of SMILE during the initial diffusion step of training. An erroneous prediction during this stage could potentially have a detrimental impact on the algorithm's performance. Initially, we considered a two-stage training approach, where we would first train the noise approximator $\epsilon_\theta$ and then train the policy $\pi_\phi$.
>
> **However, this approach turned out to be unfeasible because $\epsilon$ lacks an explicit convergence target** (in image tasks, convergence is often evaluated based on image sample-related metrics, which is not directly applicable to sequence decision-making problems). To address this challenge, we adopted an alternative strategy. While training the policy $\pi_\phi$, we additionally conducted 10 extra training steps for the $\epsilon_\theta$ in each round. This approach is commonly used in RL algorithms, such as TD3[1], to improve stability during training. Moreover, Observation 2 in Section 4.3 demonstrated the effectiveness of our method. It showed that even with potential errors of the initial predicted diffusion step in our training framework, the loss of expert demonstrations is within an acceptable range. From a technical standpoint, SMILE's stability during training can be enhanced through parameter tuning.
>
> [1] Fujimoto S, Hoof H, Meger D. Addressing function approximation error in actor-critic methods[C]//International conference on machine learning. PMLR, 2018: 1587-1596.
>
> Again, thank you for your valuable comments. Please do let us know if you have any remaining questions.
>
> Best, Authors

---

> ### Author Response · Authors · 2023-11-21
> **response to Reviewer Qmx6 Pt.4**
>
> >**Weakness 4**: The evaluations are only conducted on MuJoCo tasks. Is it able to evaluate the proposed method using one of the many existing datasets of human demos, such as RoboMimic? RoboMimic includes a classification of the level of dexterity of human demonstrations in multiple robotic tasks (in simulation), akin to the levels of noise used in the paper's experiments. Are there additional issues or limitations when applying this method to human-generated data?
>
> Following  the suggestion，we have conducted experiments in the lift and square environments within the RoboMimic framework, utilizing the low-dimension Multi-Human dataset released by RoboMimic for policy training. The experimental results are presented in the table below. Considering the short rebuttal stage and our limited computational resources, we prioritized rilco and ileed in comparison for they both handling noisy demonstrations without annotations of expertise. We evaluate all methods by showing their mean returns of 10 episode.
> |   |SMILE（ours） |  SMILE w/o filter   | rilco  | ileed |
> |  ----  | ----  |----  | ----  |----  |
> | lift | **0.7** | 0.4 | 0.0 | 0.0 |
> |square | 0.0 | 0.0 | 0.0 | 0.0 |
>
> After conducting a thorough investigation and engaging in discussions with the developers of RoboMimic, it became evident that training algorithms based on the provided dataset presented challenges in developing effective policies. It was discovered that RoboMimic might not be the most suitable task for evaluating these algorithms. The reason behind this is that the states included in the low-dimensional Multi-Human dataset from RoboMimic are not intended for training purposes, as stated by the developers themselves. These states are primarily used for resetting the simulator for specific operations like trajectory playback and observation extraction. Consequently, the states, in general, lack sufficient information to be directly utilized for policy training.
>
> Nevertheless, our observations indicate that in the "lift" task, SMILE continues to outperform other methods such as RILCO and ILEED. Based on this observation, we conjecture that there are two main factors contributing to this observation. Firstly, the dataset within RoboMimic's "lift" environment may provide relatively more informative state data compared to the "Square" environment. Secondly, SMILE is based on diffusion model which exhibits remarkable distribution fitting capabilities. Even when faced with less informative state data, SMILE is able to learn meaningful knowledge by effectively modeling the underlying distributions and capturing important patterns.
>
> Consequently, this poses a significant challenge for imitation learning algorithms as it requires meaningful representations extracted from observations. Often, this necessitates additional representation learning methods like bisimulation metrics, masking-based reconstructions, or auxiliary tasks. However, adopting such methods may introduce interference when analyzing SMILE's robustness enhancement in isolation.
>
> Therefore, we believe that experiments conducted in the Mujoco environment can better validate the effectiveness of our proposed method in a simple and intuitive way for it is a more ideal platform supporting the well-defined states,  i.e.,physical states . And the results on MuJoCo have demonstrated that SMILE is able to learn expert policy given noisy demonstrations. Though the extraction of task-relevant information from raw observations is outside the scope of our paper, we acknowledge that state feature extraction remains a pressing issue in imitation learning, and in our future work, we will continue to improve our algorithm to effectively deploy it with state feature extraction. Moreover, we have included this discussion within the 'future work' section of the main manuscript, highlighted with red fonts.

---

### Author Response · Authors · 2023-11-22
**Gentle reminder for reviewers**

We appreciate the reviewers' efforts in providing a thorough review, acknowledging the novelty of our work with its rigorous theoretical analysis and extensive empirical experiments.

We've endeavored to address all the questions and concerns raised.  However, if there are any aspects we might have inadvertently missed, we would sincerely welcome the opportunity to address them promptly.  If our responses have sufficiently clarified the concerns raised, we kindly ask the reviewers to consider re-evaluating our work in light of these clarifications and the supplementary empirical results.

---

### Meta-Review · Area_Chair_DZTR · 2023-12-05

**Metareview:**

The paper attempts to perform imitation learning form noisy demonstration. The technique is based on a diffusion model.

Strengths:
- novel, interesting approach
- new RoboMimic experiments improve the paper

Weaknesses:
- clarity of the paper is still an issue in the updated version
- reviewers have shared in the (invisible to authors) discussions stage that the theory is possibly not 100% correct (important definitions have changed in the updated version but theorems have not)
- the argument about how SAC uses multi-modal policies and how that helps dataset diversity (as included in the authors' comment) is not correct (SAC uses Gaussian policies, which are unimodal by definition).

**Justification For Why Not Higher Score:**

See weakness section in the meta-review.

**Justification For Why Not Lower Score:**

N/A

---

### Decision · Program_Chairs · 2024-01-16

Reject